# AutoRule: Reasoning Chain-of-Thought Extracted Rule-based Rewards Improve Preference Learning

## Abstract

Existing rule-based rewards in preference-based reinforcement learning rely on manual engineering, limiting scalability. We present AutoRule, a fully automated method for extracting rules from preference feedback and formulating them into rule-based rewards. AutoRule extraction operates in three stages: it leverages a reasoning model to interpret user preferences, identifies candidate rules from the reasoning chains of these interpretations, and synthesizes them into a unified rule set. Using the finalized rule set, we employ language-model verifiers to judge rule satisfaction, using this metric as an auxiliary reward alongside the learned reward model during policy optimization. Empirically, AutoRule yields gains for both Llama-3-8B and Olmo-2-7B in both in-distribution and out-of-distribution benchmarks. On Llama-3-8B, it achieves a 20.7% relative improvement in length-controlled win rate against GPT4 on AlpacaEval2.0, and a 6.1% relative gain in second-turn performance on a held-out MT-Bench subset, compared to baseline models. Further analysis shows that the extracted rules exhibit strong agreement with dataset preferences and are behaviorally consistent across multiple runs, extraction scales, and aggregated scores. Notably, these rules also contribute to mitigating reward hacking in reward models, likely because they serve as constraints that prevent the policy from exploiting spurious features. Extracted rules are provided; code and model checkpoints will be open-sourced.

## 1 Introduction

Reinforcement learning has become a cornerstone technique for aligning large language models (LLMs) with preferences and enhancing their ability to follow human instructions (Ouyang et al., 2022). RLHF and related preference-based optimization approaches have been utilized in top industry models like GPT-4 (OpenAI, 2024), Gemini (Google, 2025), Claude (Anthropic, 2024) and Llama 3 (Meta, 2024). RL-based post-training methodologies have also been leveraged to enhance the reasoning capabilities of LLMs. Notably, a key advancement in Deepseek-R1 is the adoption of rule-based rewards for accuracy and formatting in place of model-based rewards, as a strategy to mitigate reward hacking (DeepSeek-AI, 2025). Rule-based rewards for reasoning tasks are particularly effective because they provide objective, verifiable criteria that govern policy behavior. When a language model's output satisfies these rules, it can be reliably considered an accurate response.

While rule-based rewards work well for verifiable tasks, utilizing them for preference alignment in language models remains challenging. Unlike domains such as code or mathematics, where explicit rule-based verifiers are available, preference alignment is difficult because human preferences are often ambiguous and subjective. Existing industry approaches typically rely on expert-crafted rules (Glaese et al., 2022; Mu et al., 2024), large-scale crowd annotations (Bai et al., 2022b), or prompt-specific rule generations (Gunjal et al., 2025; Viswanathan et al., 2025) which can be costly and difficult to scale.

We propose AutoRule, an automatic rule-extraction framework that leverages the reasoning capabilities of frontier LLMs to derive rules directly from preference data for more subjective tasks. Here, we use the term *rule* to mean an explicit, natural-language criterion that specifies some desirable property of a response (e.g., accuracy, clarity, or adherence to instructions). Unlike prior

approaches that rely on hand-crafted, crowd-sourced, or prompt-specific rules, AUTORULE induces explicit rule-like statements from model-generated reasoning chains over a sample of preference examples, enabling broad application across the task domain. The core intuition is that each preference reflects a set of policies, and logical reasoning on these preferences and responses can reveal the underlying criteria. Aggregating these decisions across multiple examples, AUTORULE can construct a comprehensive rule set that captures the key drivers of human preferences for that particular task. During RL training, an LLM-as-a-judge verifier evaluates candidate responses for compliance with the extracted rules, and the resulting rule scores are aggregated to form a composite rule-based reward. This reward is further combined with a standard model-based reward trained to predict preferences, yielding a robust and informative signal for preference alignment.

AUTORULE consistently outperforms vanilla PPO/GRPO and other rule-based reward techniques—including simple human-crafted rules based on the UltraFeedback dimensions (Cui et al., 2024) and Rubrics as Rewards (Gunjal et al., 2025)—across multiple preference learning benchmarks. Our experiments demonstrate that rule-based scores derived from AUTORULE align closely with human preferences on both UltraFeedback and MT-Bench Human Judgment datasets, indicating the quality of the extracted rules. Importantly, AUTORULE also demonstrates strong rule consistency across runs, extraction scales, and aggregate scores, underscoring the stability and reliability of the extracted rewards. Reward hacking experiments further demonstrate that AUTORULE's rule-based rewards are robust to overoptimization, maintaining high win rates and strong generalization throughout training compared to vanilla GRPO.

In summary, our key contributions are three-fold:

- We introduce AUTORULE, a framework for automatically extracting alignment rules from preference data for subjective tasks, enabling their use as rewards in RL training.

- We show that rule-based rewards derived via AUTORULE improves preference alignment and instruction following when compared to standard preference optimization baselines.

- We demonstrate that AUTORULE produces high-quality and stable rule-based rewards that align with human preferences and enable models to mitigate reward hacking.

## 2 RELATED WORK

Reinforcement learning from human feedback (RLHF) is a standard framework for aligning large language models (LLMs) with human preferences (Ouyang et al., 2022). RLHF typically involves: (1) supervised fine-tuning on human-annotated responses; (2) training a reward model to predict human preferences; (3) reinforcement learning, commonly via proximal policy optimization (PPO) (Schulman et al., 2017) or group relative policy optimization (GRPO) (Shao et al., 2024), using a learned reward model as the optimization signal.

A well-documented challenge in RLHF with learned reward models is *reward hacking* (Bai et al., 2022a; Stiennon et al., 2022; Gao et al., 2023), in which models exploit idiosyncrasies of the reward model to achieve high reward without genuinely improving response quality. For example, Miao et al. (2024) find that reward models may overfit to superficial features, such as response length, that do not generalize to the true distribution of human preferences. Supporting this, Singhal et al. (2024) show that optimizing solely for response length during PPO can yield performance comparable to using a learned reward model, indicating that reward models often capture simple heuristics rather than more nuanced aspects of response quality.

Several strategies have been proposed to mitigate reward hacking, including modifying reward model architectures and adjusting reward scaling. ODIN (Chen et al., 2024) adds an auxiliary length prediction head to "disentangle" length from other features. Reward shaping methods such as PAR (Fu et al., 2025) and LSC (Wang et al., 2024b) apply sigmoid or log-sigmoid transformations centered on reference model outputs or percentiles. Other approaches leverage multiple reward models: WARM (Ramé et al., 2024) averages outputs from several reward models to reduce overoptimization, while ArmoRM (Wang et al., 2024a) combines interpretable reward objectives using a gating mechanism.

A growing strategy for mitigating reward hacking is the adoption of rule-based reward objectives, especially in large-scale industrial LLM deployments. For instance, DeepSeek utilized rule-based

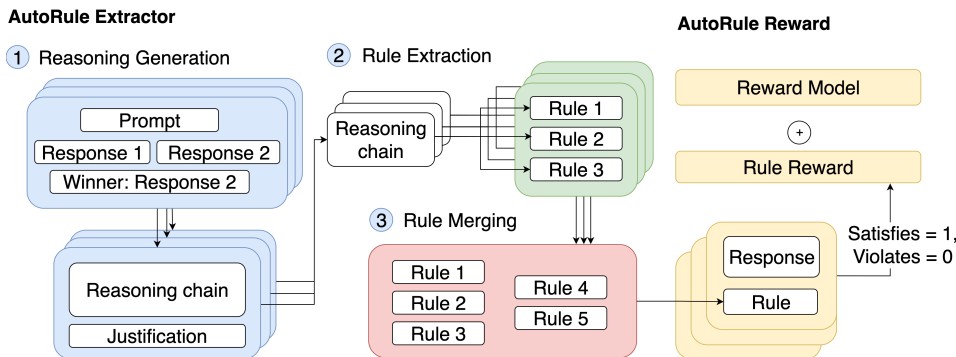

Figure 1: Overview of the AUTORULE method, which derives rule-based rewards by extracting and merging rules from reasoning chains that justify preferences.

rewards during the post-training of DeepSeek-R1 (DeepSeek-AI, 2025), explicitly prioritizing rule-based criteria over learned reward models to reduce reward hacking. Their approach incorporates two types of rewards: an accuracy reward, which evaluates whether the response is both correct and adheres to a specified format, and a format reward, which encourages the model to present its reasoning chain within designated "think" tags.

While rule-based rewards are straightforward for verifiable tasks with clear-cut success criteria (e.g., factual correctness or format adherence), extending them to preference optimization is more challenging. Human preferences are often subjective, subtle, and non-binary, making it difficult to define rules that capture the nuances of desirable behavior. Nonetheless, several works have explored rule-based objectives. Anthropic's Constitutional AI (Bai et al., 2022b) uses a curated set of constitutional principles to guide response revision and preference judgments. DeepMind's Sparrow (Glaese et al., 2022) uses researcher-defined behavioral rules, with human raters annotating violations to train a rule reward model for joint optimization with preference-based rewards. OpenAI has also investigated rule-based rewards for safety alignment, decomposing policy rules into simple propositions and using them as features in a fitted linear model to construct a reward during RL (Mu et al., 2024).

Some parallel work has explored automating rule construction by leveraging LLMs to extract per-prompt evaluation criteria, aiming to reduce the manual effort and domain expertise required for rule design. For instance, Rubrics as Rewards (RaR) (Gunjal et al., 2025) proposes generating detailed rubric items and associated importance weights for each prompt using LLMs, enabling the derivation of scalar rewards from either holistic or itemized judgments. Similarly, RLCF (Viswanathan et al., 2025) constructs prompt-specific checklists by identifying relevant failure modes and scoring responses against these criteria, which are then converted into preference data for downstream RL. These approaches represent a shift toward more systematic rule-based reward construction, though they still rely on prompt-level customization.

Although useful, constructing effective rule sets is costly, requires significant domain expertise, and often demands scenario-specific customization. Concurrent approaches, like RLCF and RaR require prompt-specific rules, which are expensive to generate and scale. As a result, rule-based approaches in preference learning remain largely proprietary within industry.

## 3 METHODS

In this section, we outline the automatic rule extraction process of AUTORULE, demonstrate how these rules can be used to form a reward score, and how the reward is used in RL. Figure 1 provides an overview of the AUTORULE pipeline. Unlike per-prompt rule generators such as RaR or RLCF, AUTORULE uses a subset of the data and forms a global rule set, achieving higher teacher efficiency.

### 3.1 AUTORULE EXTRACTOR

We denote the language model (LM) as $\pi_\theta$, where a prompt $x$ serves as the state and the next token $t$ as the action, i.e., $t \sim \pi_\theta(\cdot \mid x)$. Unrolling this process over $N$ tokens, the probability of generating

an output sequence $y = (y_1, \ldots, y_N)$ is given by $\pi_\theta(y \mid x) = \prod_{i=1}^{N} \pi_\theta(y_i \mid y_{<i}, x)$. For brevity, we write a sampled output as $y \sim \pi_\theta(\cdot \mid x)$.

The automatic rule extraction process in AUTORULE consists of three main stages, each leveraging a reasoning teacher model $\pi_\phi$ that given input $x$, decomposes a response $y$ into an output $o$ and an associated reasoning trace $r$, i.e., $(o, r) \sim \pi_\phi(\cdot \mid x)$.

**Reasoning Generation.** To guide the reasoning model toward producing a coherent, step-by-step reasoning chain suitable for rule extraction, we prompt it to justify why a selected response is chosen versus rejected. Given a preference dataset

$$\mathcal{D}_{\text{prefs}} = \left\{ (\text{instruction}^{(1)}, \text{chosen}^{(1)}, \text{rejected}^{(1)}), \ldots, (\text{instruction}^{(N)}, \text{chosen}^{(N)}, \text{rejected}^{(N)}) \right\}$$

we randomly present the reasoning model with either $\text{prompt}_{\text{reason}}(\text{instruction}, \text{chosen}, \text{rejected}, 1)$ or $\text{prompt}_{\text{reason}}(\text{instruction}, \text{chosen}, \text{rejected}, 2)$, varying the candidate order to avoid bias. The "1" and "2" represent the relative positioning of the chosen response with respect to the rejected response in the prompt. For every example $i$, we extract the reasoning trace $r^{(i)}$ from the model's generation $(o^{(i)}, r^{(i)}) \sim \pi_\phi(\cdot \mid x)$, where $x$ is the input prompt. This results in a reasoning chain collection $RC = \{r^{(1)}, \ldots, r^{(N)}\}$. The prompts used for this step and remaining steps are in Appendix J.

**Rule Extraction.** Next, we extract explicit rules from each individual reasoning chain. For every reasoning chain $r^{(i)} \in RC$, we prompt the reasoning model with $\text{prompt}_{\text{extract}}(r^{(i)})$ to elicit the underlying rules that justify the preference. The model outputs a set of rules $R^{(i)}$ for each $r^{(i)}$. We then aggregate these across all examples to obtain the overall rule set:

$$R^{(i)} \sim \pi_\phi(\cdot \mid \text{prompt}_{\text{extract}}(r^{(i)})) \qquad \mathcal{RS} = \bigcup_{i=1}^{N} R^{(i)}$$

By leveraging the reasoning model in this staged manner, we systematically decompose complex reasoning traces into precise, actionable rules. Extracting rules individually from each reasoning chain simplifies the model's task, as it avoids requiring direct rule extraction from raw preferences. This decomposition promotes higher-quality and more interpretable rule sets.

**Rule Merging.** Given the large number of rules extracted from the training set, it is crucial to merge and refine these rules to ensure computational efficiency and focus on stable, consensus-driven rules. Merging serves to reduce redundancy, eliminate noise, and consolidate semantically similar rules, thereby capturing the "wisdom of the crowd." To achieve this, we prompt the reasoning model with explicit instructions to identify and merge duplicate or overlapping rules within $\mathcal{RS}$. The resulting set is a compact, non-redundant collection of merged rules:

$$\mathcal{MR}, r \sim \pi_\phi(\cdot \mid \text{prompt}_{\text{merge}}(\mathcal{RS}))$$

where $\mathcal{MR}$ denotes the final set of merged rules. Empirically, this merging process substantially reduces redundancy and low-occurence rules, typically compressing the rule set to just 1–2% of its original size. This significantly improves the efficiency of the rule-based reward calculation process.

### 3.2 AUTORULE REWARD

To use these merged rule sets in the RL objective, we employ LLM-as-a-judge verifiers, denoted as $V_\theta$. Given a prompt $x$, a response $y$, and a rule $\text{rule}_i \in \mathcal{MR}$, the verifier produces a binary score $s_i \in \{\text{Yes}, \text{No}\}$ via $s_i \sim V_\theta(\cdot \mid \text{prompt}_{\text{verify}}(x, y, \text{rule}_i))$. A response is labeled as Yes only if it satisfies the rule and is concise. We include this conciseness requirement because reward models often exhibit undesirable length bias, and enforcing conciseness provides a principled way to mitigate it. The AUTORULE reward $r_{RA}$ is then defined as mean satisfaction across all $|\mathcal{MR}|$ rules:

$$r_{RA}(x, y) = \frac{1}{|\mathcal{MR}|} \sum_{i=1}^{|\mathcal{MR}|} \mathbb{1}\{s_i = \text{Yes}\}$$

where each $s_i$ is obtained as above. The final reward combines the rule-based reward $r_{RA}$ with the standard reward model score $r_\theta$ and the standard KL penalty (exact formulation in Appendix B.3):

$$r_{\text{total}}(x, y) = r_{RA}(x, y) + r_\theta(x, y) - \beta_{KL} KL_{\text{approx}}$$

Unlike conventional reward models which collapse preferences into a single scalar, our approach represents them as combinations of various underlying principles, so that intransitivity reflects how people differentially weight multiple dimensions of desirable behavior rather than noise. Moreover, while conventional reward models assign continuous scores reflecting subtle preference distinctions, our verifier $V_\theta$ is tasked solely producing a binary outcome on whether each rule is satisfied. This simplification reduces reward modeling complexity, making the verifier less susceptible to erroneous judgments, mitigating reward hacking risk. The rule set also enables partial rewards, offering a smooth and incremental learning signal as the policy outputs satisfy additional rules.

### 3.3 AutoRule Reinforcement Learning

While our reward system can be integrated with various policy optimization algorithms, in this work we adopt the GRPO algorithm (Shao et al., 2024) and use $r_{\text{total}}$ as the reward signal. GRPO is a policy optimization algorithm that uses the relative rewards from a group of outputs to determine advantage estimates. Formally, GRPO utilizes a group of outputs and computes their rewards, consolidating them into a reward vector $\mathbf{r} = \{r_1, \ldots, r_n\}$. Then, it computes advantage estimates for an output $i$:

$$\hat{A}_i = \frac{r_i - \text{mean}(\mathbf{r})}{\text{std}(\mathbf{r})}$$

This advantage estimate is then used in the clipped surrogate objective (Schulman et al., 2017):

$$L(w) = \mathbb{E}_{(x,y) \sim \mathcal{D}_{w_{old}}} \left[ \min \left( \frac{\pi_w(y \mid x)}{\pi_{w_{old}}(y \mid x)} \hat{A}, \text{clip} \left( \frac{\pi_w(y \mid x)}{\pi_{w_{old}}(y \mid x)}, 1 - \epsilon, 1 + \epsilon \right) \hat{A} \right) \right]$$

where $\epsilon$ is a clipping hyperparameter and $\frac{\pi_w(y|x)}{\pi_{w_{old}}}$ is the likelihood ratio.

In summary, AutoRule introduces an automated, reasoning-chain-based rule extraction framework that can generate precise and actionable alignment rules, thereby eliminating the need for manual rule engineering for subjective tasks. By decomposing rule extraction into clear, interpretable stages rather than relying on an unstable large-context single-step prompt, our approach shows that LLM reasoning chains can be systematically transformed into a coherent and effective rule set directly from preference data. Leveraging LLM-as-a-judge verifiers that provide binary rule satisfaction judgments, our approach provides additional interpretable constraints on top of conventional continuous reward models.

## 4 Experimental Methodology

**Dataset.** We use the UltraFeedback-Binarized dataset (referred to as UltraFeedback), a binarized version of UltraFeedback (Cui et al., 2024), which contains nearly 64K pairwise preference annotations across diverse model types and instructions. This dataset provides a large-scale and heterogeneous base, making it well-suited for evaluating generalization across prompts of varying difficulty. For training, we select a filtered subset of 33K examples (details in Appendix B.6). In addition, we leverage the MT-Bench human judgment dataset (Mu et al., 2024), which provides expert preference annotations on multi-turn questions.

**Evaluation Metrics.** We report win rate on the UltraFeedback-Binarized test split, using GPT-4o as an automatic judge with randomized candidate and reference response order. We also evaluate on MT-Bench (using a GPT-4 judge) and AlpacaEval 2.0 (Dubois et al., 2024). AlpacaEval complements UltraFeedback and MT-Bench by emphasizing instruction-following in a single-turn setting, while also correcting for verbosity bias via its length-controlled win-rate. For AutoRule, AlpacaEval 2.0 and UltraFeedback win rate are measured on a model trained with rules from UltraFeedback. For MT-Bench, we split the 80 questions in half for training/testing (5 per category for each split).

Together, these benchmarks span a spectrum of evaluation regimes—ranging from single-turn instruction following (AlpacaEval 2.0), to multi-turn dialogue across diverse categories (MT-Bench), to large-scale, heterogeneous preference data (UltraFeedback).

**Rule Extraction.** We use Deepseek-R1 (DeepSeek-AI, 2025) to generate reasoning chains for automatic rule extraction. For the LLM-as-a-judge verifier, we use Llama-3-8B-Instruct (Meta, 2024)

Table 1: Main evaluation results on UltraFeedback win-rate vs SFT, AlpacaEval 2.0 (denoted as "AE") length-controlled and vanilla win-rate, and MT-Bench average, turn 1, and turn 2 score. *Variant w/o code, strongest reported performance on conversational assistance benchmarks.

| | OLMo-2-7B (base) | | | | Llama-3-8B (base) | | | |
| | UF | AE LC WR (WR) | | MT-Bench | UF | AE LC WR (WR) | | MT-Bench |
| **Methods** | **WR** | **vs SFT** | **vs GPT4** | **Avg (T1/T2)** | **WR** | **vs SFT** | **vs GPT4** | **Avg (T1/T2)** |
|---|---|---|---|---|---|---|---|---|
| SFT | – | – | $6.6^{(4.7)}$ | $6.05^{(6.75/6.35)}$ | – | – | $10.8^{(7.2)}$ | $6.40^{(6.98/5.83)}$ |
| PPO | 74.6 | $72.8^{(74.6)}$ | $14.0^{(10.9)}$ | $6.79^{(7.15/6.43)}$ | 67.6 | $66.3^{(67.0)}$ | $15.2^{(11.1)}$ | $7.09^{(7.41/6.78)}$ |
| GRPO | 76.6 | $76.4^{(79.4)}$ | $15.7^{(12.8)}$ | $6.75^{(7.16/6.33)}$ | 75.9 | $72.7^{(82.2)}$ | $15.1^{(16.1)}$ | $7.68^{(7.98/7.38)}$ |
| + Length Control | 75.5 | $76.8^{(80.1)}$ | $15.6^{(12.6)}$ | $6.91^{(7.21/6.60)}$ | 75.9 | $66.1^{(80.2)}$ | $16.8^{(16.8)}$ | $7.40^{(7.45/7.35)}$ |
| + Length Penalty | 74.6 | $77.2^{(77.4)}$ | $15.7^{(10.6)}$ | $6.60^{(6.95/6.25)}$ | 76.1 | $71.0^{(76.6)}$ | $16.2^{(12.5)}$ | $7.29^{(7.58/7.00)}$ |
| + Concise Rule | 72.8 | $77.5^{(75.6)}$ | $18.6^{(11.1)}$ | $7.01^{(7.51/6.51)}$ | 69.9 | $71.6^{(75.8)}$ | $17.2^{(13.7)}$ | $7.06^{(7.49/6.63)}$ |
| RaR-Implicit | 66.2 | $58.3^{(63.5)}$ | $9.8^{(8.8)}$ | $6.62^{(7.24/6.00)}$ | 63.2 | $59.4^{(66.0)}$ | $13.6^{(11.5)}$ | $7.11^{(7.41/6.80)}$ |
| RaR-Explicit | 70.3 | $68.9^{(77.9)}$ | $13.0^{(14.1)}$ | $6.68^{(7.30/6.05)}$ | 70.4 | $64.6^{(78.0)}$ | $13.0^{(14.1)}$ | $7.06^{(7.33/6.80)}$ |
| RLCF* | 71.4 | $67.9^{(80.1)}$ | $13.1^{(18.5)}$ | $7.01^{(7.53/6.50)}$ | 71.3 | $66.4^{(83.7)}$ | $17.9^{(25.0)}$ | $7.34^{(7.90/6.78)}$ |
| AUTORULE | **79.4** | $\mathbf{81.6}^{(83.9)}$ | $\mathbf{20.0}^{(15.9)}$ | $\mathbf{7.03}^{(7.25/6.80)}$ | **77.2** | $\mathbf{77.0}^{(83.3)}$ | $\mathbf{21.6}^{(18.6)}$ | $\mathbf{7.85}^{(7.88/7.83)}$ |

for computational efficiency. To extract rules, we sample 256 random examples from the Ultra-Feedback training split and for MT-Bench, we use the 40-question training split and sample up to 8 examples per question for training, or all available if fewer.

**Baselines.** We first compare to the SFT checkpoint ("SFT") and a PPO baseline. Next, we consider GRPO-based baselines: (1) GRPO with the base reward ($r_\theta$) ("GRPO"), (2) GRPO with length-driven hyperparameter tuning ("GRPO + Length Control", LC), (3) GRPO with a length penalty ("GRPO + Length Penalty", LP), and (4) GRPO with a single concise rule reward ("GRPO + Concise Rule"); all use the same learned reward model. Finally, we evaluate parallel methods with GRPO: (5) RaR-Implicit, which uses a holistic 1–10 rubric judge as reward, (6) RaR-Explicit, which aggregates per-item rubric checks by a normalized weighted sum, and (7) RLCF, which similarly aggregates per-item checks. Rule-based baseline details are in Appendix B.7.

**AUTORULE Model.** For AUTORULE, we use a scaled rule-based reward $r_{RA}$: $r_{RA'} = \alpha r_{RA} + \beta$, aligning the rule-based reward magnitude with the learned reward model for stable training. The verifier prompt is modified so $s_i = 1$ only if the response is concise and fully satisfies the extracted rule. We set $(\alpha, \beta) = (10, -7.5)$ for AUTORULE on Llama-3-8B (UF and MT rules) and OLMo-2-7B (UF rules), and $(\alpha, \beta) = (5, -3)$ for OLMo-2-7B (MT rules).

**Implementation Details.** All models are initialized from the same SFT and reward model checkpoints for comparability. SFT checkpoints are obtained by fine-tuning Llama-3-8B and OLMo-2-7B (OLMo et al., 2025) on chosen responses from filtered UltraFeedback-Binarized. The reward model is initialized from this SFT checkpoint and further fine-tuned on filtered UltraFeedback-Binarized preferences. Actor, critic, and value networks are initialized from the SFT checkpoint. Training uses OpenRLHF (Hu et al., 2024), an open-source RLHF framework, of which we modified to support LLM-judged rewards. Training details and asset URLs are in Appendix B and K, respectively.

## 5 EVALUATION RESULTS

In this section, we present a comprehensive evaluation of AUTORULE by analyzing model performance, teacher model efficiency, ablation studies, rule effectiveness, and reward hacking mitigation.

### 5.1 MODEL PERFORMANCE

Table 1 demonstrates that AUTORULE achieves strong in-distribution performance on both Ultra-Feedback and MT-Bench. For Llama-3-8B, AUTORULE delivers a 1.4% relative improvement in UltraFeedback win rate and a 6.1% relative gain in MT-Bench Turn 2 performance over the best

baseline, highlighting the effectiveness of rule-based rewards in capturing human preferences and supporting complex, multi-turn interactions.

Furthermore, AUTORULE demonstrates strong out-of-distribution generalization and robustness to length bias. On AlpacaEval 2.0, AUTORULE, using rules extracted from UltraFeedback data, achieves a 5.9% relative improvement in length-controlled win rate against SFT and a 20.7% improvement against GPT-4 Turbo over the best baseline, indicating that rule-based rewards promote substantive response quality rather than superficial length-based cues. These gains also generalize across different LLM families. For OLMo-2-7B, AUTORULE attains the highest UltraFeedback win, AlpacaEval 2.0 length-controlled win rate, best average MT-Bench score, among all methods.

Collectively, these results show that AUTORULE not only excels within its training distribution but also transfers effectively to diverse evaluation settings and model families.

**Teacher Model Efficiency.** Unlike concurrent approaches such as RaR and RLCF, which necessitate teacher model inference for every training prompt, AUTORULE requires only a subset of examples to extract its rule set (256 vs full 33K training set). As shown in Table 2, AUTORULE results in magnitudes lower teacher model token consumption. This underscores the practicality of AUTORULE.

Table 2: Teacher model tokens. *Estimated token count extrapolated from 256 examples.

| Method | Input | Output | Total |
|---|---|---|---|
| AUTORULE | 490K | 493K | 983K |
| RaR* | 34.2M | 54.5M | 88.7M |
| RLCF | 147.1M | 56.0M | 203.1M |

## 5.2 ABLATION STUDY

We perform ablations to analyze contributions of teacher model, rule source, extraction source, rule count, rule reward construction, and judge type.

**Teacher Model.** We conducted an additional experiment using Qwen3-8B as the teacher model for the AutoRule extraction pipeline, with results shown in Table 3. The results demonstrate that the rule-extraction process remains effective even with a smaller teacher model, yielding downstream performance comparable to DeepSeek-R1.

**Rule Source.** Designing effective rules manually remains a challenge. When we employ the text descriptions of the four UltraFeedback dimensions (with "LLM" replaced by "The assistant") as rules, performance is notably lower than AUTORULE (as shown in Table 3. To better understand this gap, we analyze the optimization dynamics on a per-rule level. Table 4 presents per-rule scores at early and late training (step 1 vs. 32), as well as preference alignment. Notably, three of the UltraFeedback dimension rules are nearly saturated, which forces op-

Table 3: Ablation study of teacher model, rule source, and extraction source on Ultra-Feedback win-rate vs SFT and AlpacaEval 2.0 (AE) length-controlled and vanilla win-rate. All methods use Llama-3-8B as the base model.

| Method | UF WR | AE LC WR (WR) vs SFT | vs GPT4 |
|---|---|---|---|
| Best Baseline | 76.1 | 72.7 [82.2] | 16.8 [16.8] |
| *Teacher Model* | | | |
| DeepSeek-R1 | 77.2 | **77.0** [83.3] | 21.6 [18.6] |
| Qwen3-8B | **78.5** | 76.1 [82.3] | **22.0** [18.6] |
| *Rule Source* | | | |
| AUTORULE | 77.2 | **77.0** [83.3] | **21.6** [18.6] |
| UF Dimensions | 54.7 | 46.4 [61.4] | 8.7 [9.3] |
| *Extraction Source* | | | |
| Reasoning | 77.2 | **77.0** [83.3] | **21.6** [18.6] |
| Justification | 75.9 | 75.9 [82.2] | 19.7 [16.5] |

timization onto a single signal (the fourth rule). Qualitative analysis further reveals that these human-authored rules are relatively vague. In contrast, the rules automatically extracted by AUTORULE are more specific, as they are derived directly from preference data. We observe that many of these rules exhibit meaningful score increases after training (see Appendix Figure 10).

**Extraction Source.** We find that extracting rules from the reasoning chain, rather than from the final justifications, yields substantially higher UltraFeedback and AlpacaEval 2.0 length-controlled win rates (Table 3). This suggests that reasoning chains provide more specific and actionable guidance for rule formulation, whereas justifications tend to be less detailed and more generic, leading to diminished downstream performance. We further analyze this with a case study in Appendix F.

Table 4: Per-rule training dynamics (step $1 \to 32$) and preference alignment.

| Rule | $\Delta$ Score | Align |
|---|---|---|
| The assistant should respond to humans without deviating from the requirements. | $0.91 \to 0.96$ | 84.6% |
| The assistant should provide useful and correct answers to address the given problems. | $0.91 \to 0.96$ | 87.1% |
| The assistant's output should be grounded in the instructions and real-world knowledge, and avoid introducing any self-contradiction. | $0.92 \to 0.96$ | 87.2% |
| The assistant should know what they (don't) know and express uncertainty towards the given problem. | $0.37 \to 0.81$ | 64.4% |

**Rule Count.** We evaluated 5-rule and 50-rule variants (in addition to the original 25-rule set) to analyze the scaling behavior of the merged rule set. The results, shown in Table 5, show that AutoRule's downstream performance is sensitive to the number of extracted rules. While AlpacaEval metrics remain relatively stable across configurations, the UF win-rate varies more noticeably. We hypothesize that the 5-rule configuration contains too few applicable rules to provide sufficiently detailed guidance, whereas the 50-rule configuration introduces many rules that are rarely applicable, diluting the reward signal. The 25-rule setting strikes a stronger balance, offering enough coverage without excessive noise.

**Rule Reward Construction.** Both reward scaling and the explicit inclusion of conciseness as a reference in the verifier prompt are critical for optimal performance. To assess their impact, we evaluate two ablation variants: (1) a version without con-

Table 5: Ablation study of rule count and rule reward construction on UltraFeedback winrate vs SFT and AlpacaEval 2.0 (AE) length-controlled and vanilla win-rate. All methods use Llama-3-8B as the base model.

| | UF | AE LC WR (WR) | |
|---|---|---|---|
| **Method** | **WR** | **vs SFT** | **vs GPT4** |
| Best Baseline | 76.1 | 72.7 $^{(82.2)}$ | 16.8 $^{(16.8)}$ |
| *Rule Count* | | | |
| 5 rules | 69.3 | 75.7 $^{(76.1)}$ | 21.0 $^{(13.6)}$ |
| 25 rules | **77.2** | 77.0 $^{(83.3)}$ | 21.6 $^{(18.6)}$ |
| 50 rules | 74.9 | **77.1** $^{(83.0)}$ | **23.5** $^{(19.9)}$ |
| *Rule Reward Construction* | | | |
| Scale+Concise | **77.2** | **77.0** $^{(83.3)}$ | **21.6** $^{(18.6)}$ |
| Scale+NoConcise | 74.6 | 65.2 $^{(82.4)}$ | 16.5 $^{(21.6)}$ |
| NoScale+NoConcise | 75.7 | 68.6 $^{(82.5)}$ | 14.5 $^{(17.8)}$ |

ciseness references ("Scale+NoConcise"), and (2) a version with reward scaling parameters set to $\alpha = 1, \beta = 0$ and no conciseness references ("NoScale+NoConcise"). Results for these variants, alongside the original methåod ("Scale+Concise"), are shown in the lower section of Table 5. We observe that removing either reward scaling or conciseness guidance consistently reduces both UltraFeedback win rate and AlpacaEval 2.0 length-controlled win rates. The lack of reward scaling appears to limit the model's ability to fully leverage rule-based supervision, while omitting conciseness guidance leads to responses that are less aligned with human preferences for brevity and clarity. These findings highlight the necessity of both rule reward scaling and conciseness encouragement within the AUTORULE framework.

Note that conciseness alone does not account for the observed performance improvements. As shown by the "GRPO + Concise Rule" baseline in Table 1, training with only a single conciseness rule yields substantially lower performance than AUTORULE.

**Judge Type.** Next, we analyze the importance of judge type and whether the performance gain arises from judge compute itself during reward computation. Due to the 8k context-length limit of the judge model, we cannot fully match the compute of the original 25-rule setting using reasoning. Instead, we evaluate two alternatives: (1) a single LLM judge call prompted with a detailed reasoning guide, inducing $\sim$1.5K tokens of thinking and (2) averaging 3 calls with smaller reasoning chains ($\sim$ 300-tokens) which align more closely with the model's natural chain-of-thought length. The results, displayed in Table 6, demonstrate that al-

Table 6: Judge-type comparison on UF winrate, AlpacaEval 2.0 win-rates, and estimated FLOPs per reward score. AR is AUTORULE.

| | UF | AE LC WR (WR) | | **FLOPs** |
|---|---|---|---|---|
| **Method** | **WR** | **vs SFT** | **vs GPT4** | **/ score** |
| AR n=25 | **77.2** | **77.0** $^{(83.3)}$ | **21.6** $^{(18.6)}$ | 200.4 T |
| AR n=5 | 69.3 | 75.7 $^{(76.1)}$ | 21.0 $^{(13.6)}$ | **40.1 T** |
| Long n=1 | 61.9 | 53.6 $^{(60.5)}$ | 11.6 $^{(9.9)}$ | 46.0 T |
| Short n=3 | 64.0 | 60.7 $^{(79.0)}$ | 16.9 $^{(27.9)}$ | 42.7 T |

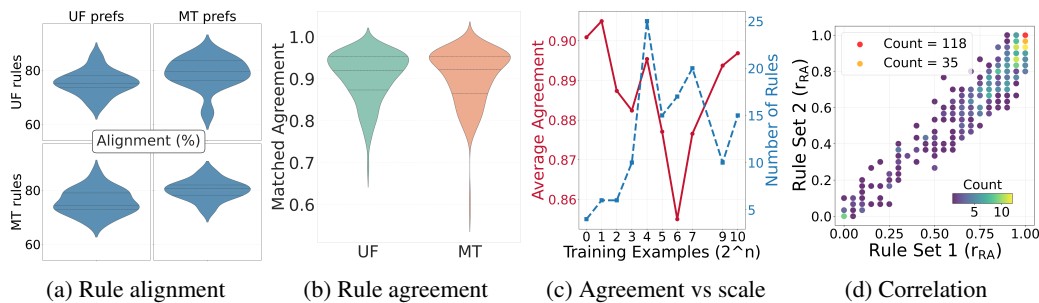

(a) Rule alignment  (b) Rule agreement  (c) Agreement vs scale  (d) Correlation

Figure 2: 2a shows rule alignment distributions for UF and MT-extracted rules, 2b plots matched rule behavior agreement, 2c shows average behaviorial agreement between rules at different training example scales matched with the original rule set (extracted from 256 examples), and 2d plots rule scores for two AUTORULE runs.

though AUTORULE (5 rules) performs worse in win rate for UltraFeedback compared to the original AUTORULE setting (25 rules), both LLM-judge baselines perform worse than the 5-rule setting in both benchmarks, despite using comparable compute. The short-thought ensemble in particular exhibits clear length hacking signs. These findings suggest that AUTORULE's improvements do not stem from compute alone, but rather come from the structure and granularity of rule-based feedback.

## 5.3 EFFECTIVENESS OF AUTORULE RULES

In the following experiments, we analyze the effectiveness of AUTORULE to extract high-quality, consistent rules. All extracted rules are displayed in Appendix C, and a case study comparing rules extracted from different datasets is provided in Appendix G.

**Rule quality.**  To assess rule quality, we calculate rule alignment on 1,024 UltraFeedback test examples and the full MT-Bench human judgment split. Alignment is computed as

$$\text{Alignment} = \frac{\sum_{\text{pairs}} \mathbf{1}\{s_{\text{chosen}} = 1 \land s_{\text{rejected}} = 0\}}{\sum_{\text{pairs}} \mathbf{1}\{s_{\text{chosen}} \neq s_{\text{rejected}}\}}$$

Figure 2a presents the distributions for individual rule alignment. We observe that individual rules from both rule sets are in good alignment with the ground-truth dataset preferences. The observed alignment also indicates that the extracted rules generalize effectively across datasets, demonstrating that they capture accurate and transferable decision principles. Finally, we observe higher alignment overall when evaluating against MT preferences, likely due to the expert-level annotations used.

**Rule consistency.**  To assess the stability of individual rules, we construct a binary vector for each rule, indicating whether the rule is satisfied for 512 responses, and compute the normalized Hamming distance as a measure of *behavioral agreement*. To compare two separate rule sets for consistency, we match rules using the Hungarian algorithm (Kuhn, 1955), and report the behavioral agreement of matched pairs. The resulting distributions, shown in Figure 2b, reveal that the mean matched agreement for pairwise rule sets across 5 runs exceeds 0.9 for both datasets, demonstrating that AutoRule consistently extracts highly stable rules.

We further examine the robustness of rule consistency with respect to extraction scale. Specifically, we compare rules extracted from varying numbers of examples to those obtained from the original set of 256 examples, again using behavioral agreement and the Hungarian algorithm for matching. As illustrated in Figure 2c, while the number of merged rules decreases as $n$ is reduced, the average behavioral agreement remains high. This suggests that, although fewer examples may yield a less comprehensive rule set, the extracted rules themselves remain consistent. Collectively, these results highlight that rule extraction via AUTORULE produces robust rules across different extraction scales.

Additionally, we assess aggregate rule score consistency by analyzing the correlation between aggregated scores assigned by different rule sets, as shown in Figure 2d. The mean pairwise correlations across five runs are 0.967 for UltraFeedback and 0.964 for MT-Bench, further confirming the strong consistency and reproducibility of the AUTORULE extraction process in generating reliable aggregate scores. Additional detailed consistency experiments are presented in Appendix E.

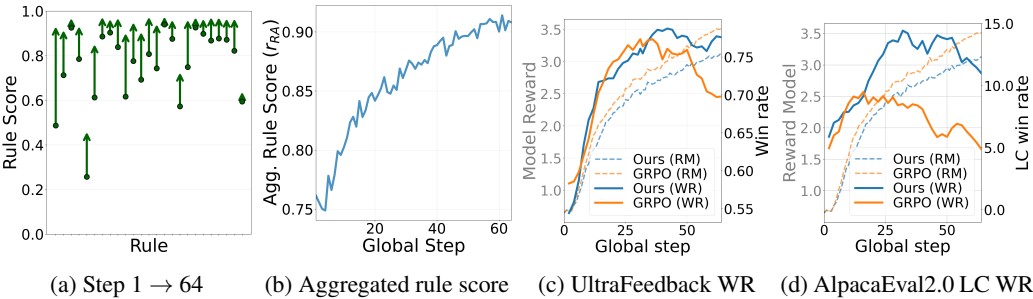

(a) Step 1 → 64  (b) Aggregated rule score  (c) UltraFeedback WR  (d) AlpacaEval2.0 LC WR

Figure 3: 3a and 3b show upward rule score trends, and 3c and 3d show 128-example subset evals of AUTORULE and GRPO (Llama-3-8B) for two episodes with 10-step rolling average smoothing.

Finally, we conducted an experiment evaluating how rule-set size affects alignment accuracy. For rule sets of various sizes, we report how often the aggregated rule score assigns a higher, equal, or lower score to the chosen response. The results, displayed in Table 7, show a large improvement from 5 → 25 rules, indicating that a sufficient number of rules is needed to meaningfully differentiate preferences. However, the gain from 25 to 50 rules is marginal, with diminishing returns and slightly more contradictory signals (higher < 0 count).

Table 7: Distribution of preferred response aggregated score margins across different rule counts for 256 UF examples.

| Rules | $> 0\,(\%)$ | $= 0\,(\%)$ | $< 0\,(\%)$ |
|---|---|---|---|
| 5 | 43.8 | 34.4 | 21.9 |
| 25 | 59.4 | 11.7 | 28.9 |
| 50 | 62.1 | 5.5 | 32.4 |

### 5.4 REWARD HACKING MITIGATION

Following prior work (Miao et al., 2024; Fu et al., 2025), a standard way to detect reward hacking is to track a true (gold) metric over the course of training and observe whether it initially increases and then later declines. Our results show that AUTORULE helps prevent reward hacking on in-distribution benchmarks. Figure 3c plots the UltraFeedback win rate as a function of global step, smoothed with a rolling average over 10 steps. Initially, both the baseline and AUTORULE models achieve similar win rates; however, after step 52, the GRPO baseline exhibits declining performance, whereas AUTORULE maintains consistently high win rates. This illustrates that the baseline is "hacking" the reward by exploiting spurious local optima where the reward increases but the true win rate decreases, whereas both metrics continue to improve for AUTORULE. We analyze an illustrative example in Appendix H.

The AUTORULE approach also mitigates reward hacking on out-of-distribution benchmarks. Figure 3d shows the AlpacaEval 2.0 win rate as a function of global step, also smoothed with an 10-step rolling average. Here, AUTORULE consistently outperforms GRPO, achieving an improvement of roughly 5% after two episodes. These results demonstrate that rule-based rewards provide robustness against reward hacking in both in-distribution and out-of-distribution settings.

We attribute this robustness to the effectiveness of rule-based rewards. As shown in Figures 3a and 3b, the consistent upward trend in rule scores indicates that the AUTORULE model reliably optimizes the rule-based reward signal. The underlying intuition is that rule-based rewards serve as constraints, limiting the model's ability to exploit weaknesses or spurious local optima in the reward model.

## 6 CONCLUSION

In this paper, we introduce AUTORULE, a reasoning chain-based automatic rule extraction mechanism for leveraging rule-based rewards in language model alignment. We demonstrate that rules extracted by AUTORULE align well with preference datasets and improve performance on instruction-following benchmarks. Additionally, we show that rule-based rewards help mitigate certain aspects of reward model overoptimization. We hope that AUTORULE will enable AI researchers and practitioners to construct and utilize more effective rule-based rewards in post-training pipelines, ultimately advancing the development of models for creative and subjective tasks.

## 7 REPRODUCIBILITY STATEMENT

An overview of our experimental setup is provided in Section 4. Detailed descriptions of training procedures, including data filtering, inference and training hyperparameters, and KL divergence approximations, are presented in Appendix B. The complete codebase will also be released to facilitate reproducibility.

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

# A  LLM USAGE

We describe the ways in which LLMs were utilized in this work:

**Polishing writing.** LLMs were employed to refine language, grammar, and overall clarity throughout the paper.

**Retrieval and discovery.** LLMs assisted in identifying related work, including research on length-based reward hacking and rule-based objectives for alignment.

**Research ideation.** LLMs contributed to the selection and formulation of certain evaluation metrics, such as the AlpacaEval 2.0 win rate and MT-Bench score, as well as to the choice of the datasets for rule extraction and model training.

**Other.** LLMs are central to this work itself, serving as the models being aligned, as judges for evaluation, and as mechansism for rule generation.

# B  ADDITIONAL EXPERIMENT DETAILS

## B.1  TRAINING SETTINGS

Settings used for the SFT, reward model, and RL training are available in Tables 8, 9, and 10 respectively.

## B.2  INFERENCE PARAMETERS

Inference parameters are displayed in Table 11.

## B.3  KL APPROXIMATION

We utilize two versions of KL approximation as implemented in OpenRLHF (Hu et al., 2024). The first is used for PPO, and the second is used for GRPO.

$$\log \left( \frac{\pi_\phi(y \mid x)}{\pi^{SFT}(y \mid x)} \right) \tag{1}$$

$$e^{-\log\left( \frac{\pi_\phi(y|x)}{\pi^{SFT}(y|x)} \right)} - 1 + \log \left( \frac{\pi_\phi(y \mid x)}{\pi^{SFT}(y \mid x)} \right) \tag{2}$$

## B.4  LENGTH PENALTY

To implement the length penalty, we subtract the following from the reward:

$$\frac{1}{2} \left( \frac{\text{response\_length}}{L} \right) - \frac{1}{2}$$

where $L = 300$ is the target length.

## B.5  GRPO ADVANTAGE ESTIMATION

To improve numerical stability, as implemented in OpenRLHF, we utilize a modified version of the advantage estimation formula displayed in Section 3.3 as follows:

$$\hat{A}_i = \frac{r_i - \text{mean}(\mathbf{r})}{\text{std}(\mathbf{r}) + 10^{-9}}$$

## B.6  DATASET FILTERING

Following the filtering process and using the code by (Fu et al., 2025), to select data for training, we filter and only include the examples where the chosen and rejected responses are both less than 512 tokens, the chosen score is higher than the rejected score, and the word "confidence" is not in the either response.

## B.7 RULE-BASED BASELINES

We evaluate three rule-based baselines that incorporate auxiliary rule rewards into GRPO.

**GRPO + Concise Rule.** This baseline applies a single auxiliary rule that encourages brevity: "The assistant should respond in a concise manner." Compliance is assessed using the same concise verifier prompt used in AUTORULE (Figure 14).

**RaR.** We employ Deepseek-R1 as a teacher model to synthesize rubrics for each UltraFeedback example, following Gunjal et al. (2025) with prompt adaptations for instruction-following and alignment tasks. We study two variants: (1) GRPO + RaR-Implicit, which uses the rubric's holistic 1–10 score as the reward; and (2) GRPO + RaR-Explicit, which composes an explicit reward by aggregating normalized per-item checks via a weighted sum.

**RLCF.** We employ Deepseek-R1 as a teacher model to synthesize checklists for each UltraFeedback example, following Viswanathan et al. (2025) for the checklist synthesis process.

All rule-based baselines use the same GRPO hyperparameters as the "GRPO" model, which are detailed in Table 10.

Table 8: Supervised fine-tuning settings.

| Setting | Value |
|---|---|
| Base model | Llama-3-8B/OLMo-2-7B |
| Dataset / split | UltraFeedback SFT training split |
| Epochs | 2 |
| Train / micro batch | 256 / 2 |
| Learning rate | $5 \times 10^{-6}$ |
| LR scheduler | Constant with warmup |
| LR warmup ratio | 0.1 |
| Adam $\beta_1, \beta_2$ | 0.9, 0.95 |
| Precision | bfloat16 |
| Grad-norm clip | 10 |
| Seed | 42 |

Table 9: Reward model training settings.

| Setting | Value |
|---|---|
| Base checkpoint | SFT checkpoint |
| Loss | Pairwise sigmoid |
| Epochs | 1 |
| Train / micro batch | 256 / 2 |
| Learning rate | $5 \times 10^{-6}$ |
| LR scheduler | Constant with warmup |
| LR warmup ratio | 0.1 |
| Adam $\beta_1, \beta_2$ | 0.9, 0.95 |
| Attention | Flash |
| Precision | bfloat16 |
| Grad-norm clip | 5 |
| Seed | 42 |

Table 10: RL training settings. AUTORULE and GRPO + LP uses the same settings as GRPO, except the verifier for AUTORULE uses a temperature of 0.0.

| Setting | PPO | GRPO | GRPO + LC |
|---|---|---|---|
| Actor init (policy) | SFT ckpt | SFT ckpt | SFT ckpt |
| Reward / critic init | RM ckpt | RM ckpt | RM ckpt |
| Dataset / split | UF Prefs training split | same | same |
| KL estimator version | (1) | (2) | (2) |
| Initial $\beta_{\mathrm{KL}}$ | 0.005 | 0.001 | 0.005 |
| $\lambda$ | 0.95 | 1.00 | 1.00 |
| $\gamma$ | 1 | 1 | 1 |
| Samples per prompt | 1 | 2 | 2 |
| PTX coefficient | 0.05 | 0.05 | 0.05 |
| Actor learning rate | $3 \times 10^{-7}$ | $5 \times 10^{-7}$ | $3 \times 10^{-7}$ |
| Critic learning rate | $5 \times 10^{-6}$ | $9 \times 10^{-6}$ | $9 \times 10^{-6}$ |
| LR scheduler | Constant w/ warmup | same | same |
| LR warmup ratio | 0.1 | 0.1 | 0.1 |
| Adam $\beta_1, \beta_2$ | 0.9, 0.95 | 0.9, 0.95 | 0.9, 0.95 |
| Rollout batch (total/micro) | 1024 / 32 | 1024 / 32 | 1024 / 32 |
| Train batch (total/micro) | 128 / 16 | 128 / 16 | 128 / 16 |
| Temperature / top-$p$ | 0.9 / 0.9 | 1.0 / 1.0 | 0.9 / 1.0 |
| vLLM max new tokens | 1024 | 1024 | 1024 |
| Epochs | 1 | 1 | 1 |
| Precision | bfloat16 | bfloat16 | bfloat16 |
| Attention | Flash | Flash | Flash |
| Seed | 42 | 42 | 42 |
| Grad-norm clip | 5 | 1 | 1 |

Table 11: Inference parameters used, "preset" means it is defined by the benchmark.

| Model (Purpose) | Temperature | Top P | Max new tokens |
|---|---|---|---|
| Verifier (Rule agreement experiments) | 0.0 | 1.0 | 256 |
| Verifier (Determinism experiment) | 1.0 | 1.0 | 256 |
| Trained model (UF win-rate) | 0.9 | 1.0 | 1024 |
| Trained model (AlpacaEval 2.0) | 0.9 | 1.0 | 1024 |
| Trained model (MT-Bench) | Preset | 1.0 | 1024 |
| Deepseek-R1 (Full rule extraction process) | 0.6 | 1.0 | 32768 |

## C  RULES

Rules extracted from UltraFeedback (reasoning chain), MT-Bench (reasoning chain), and Ultra-Feedback (justification) are displayed in Tables 12, 13, 14 respectively.

Table 12: UltraFeedback rules extracted via the AUTORULE extraction process.

| Index | Rule | Align (%) |
|---|---|---|
| 0 | The assistant's responses should present explanations in a coherent, step-by-step structure with logical flow, numbered points, and clear sections. | 75.1 |
| 1 | When addressing user misconceptions, the assistant must clarify misunderstandings before offering solutions. | 75.9 |
| 2 | Translations must use accurate terminology, preserve original tone and structure, and avoid introducing unrelated content. | 79.4 |
| 3 | Responses must prioritize technical accuracy, correct formulas, error-free code examples, and validated context alignment. | 76.2 |
| 4 | Incorporate vivid sensory details, figurative language, and relatable examples when explicitly requested. | 74.1 |
| 5 | Provide actionable advice, practical steps, and concrete implementation strategies tailored to the user's context. | 74.8 |
| 6 | Indicate confidence levels while acknowledging uncertainty and limitations when appropriate. | 74.8 |
| 7 | Maintain a conversational, empathetic, and professional tone while avoiding overly formal or dismissive language. | 71.4 |
| 8 | Integrate cultural sensitivity, domain-specific terminology, and contextual relevance into explanations. | 73.9 |
| 9 | Include properly formatted citations, references, and academic conventions when required. | 73.1 |
| 10 | Address all components of the user's query comprehensively without omission or tangential content. | 73.6 |
| 11 | Avoid assumptions when ambiguity exists; seek clarification for insufficient context. | 69.9 |
| 12 | Use illustrative examples of both correct/incorrect approaches to demonstrate concepts. | 78.2 |
| 13 | Strictly adhere to user-specified formats, structures, and output requirements. | 70.2 |
| 14 | Address ethical considerations, legal compliance, and recommend professional consultation when relevant. | 80.9 |
| 15 | Prioritize security measures, error handling, and technical robustness in solutions. | 78.1 |
| 16 | Ensure conciseness by eliminating redundancy and focusing on core query relevance. | 67.4 |
| 17 | Explain underlying mechanisms, reasoning processes, and cause-effect relationships explicitly. | 74.8 |
| 18 | Validate answers against provided context and avoid unsupported extrapolation. | 85.5 |
| 19 | Maintain narrative coherence with source material when discussing plots or characters. | 84.3 |
| 20 | Structure comparisons, analyses, and recommendations using clear categorization. | 76.1 |
| 21 | Anticipate user needs by providing comprehensive details without requiring follow-ups. | 78.6 |
| 22 | Preserve specific terms, measurements, and formatting conventions during localization. | 76.3 |
| 23 | Use collaborative language and hierarchical organization for complex information. | 77.4 |
| 24 | Balance thoroughness with brevity to prevent information overload while ensuring clarity. | 66.6 |

Table 13: MT-Bench rules extracted via the AUTORULE extraction process.

| Index | Rule | Align (%) |
|:---:|:---|:---:|
| 0 | The assistant's responses must provide detailed step-by-step explanations and calculations to ensure correctness and clarity. | 86.7 |
| 1 | The assistant's code should avoid unnecessary complexity, handle edge cases, include error handling, and use appropriate data structures. | 80.4 |
| 2 | The assistant's responses must maintain a professional and approachable tone, adapting to the nature of the user's query. | 83.8 |
| 3 | The assistant's responses must strictly adhere to user-specified formats (e.g., JSON/YAML) with correct syntax and structure. | 72.2 |
| 4 | The assistant's explanations should prioritize logical coherence, clarity, and avoidance of redundant or ambiguous content. | 78.2 |
| 5 | The assistant must adhere to ethical guidelines by avoiding medical diagnoses and prioritizing user safety in critical situations. | 77.0 |
| 6 | Creative outputs must maintain structural integrity (e.g., rhyme schemes, metaphors) while retaining key informational elements. | 78.7 |
| 7 | The assistant should proactively address user misunderstandings, anticipate follow-up questions, and provide actionable feedback. | 86.7 |
| 8 | The assistant must apply appropriate theoretical principles (e.g., Bayes' theorem) and clarify their relevance to the problem. | 81.7 |
| 9 | The assistant's responses should validate assumptions, acknowledge limitations, and use verified data in calculations. | 77.2 |
| 10 | The assistant must tailor recommendations to user constraints (e.g., allergies, pregnancy) and cultural context. | 81.6 |
| 11 | The assistant's structured outputs should prioritize readability through proper formatting and organizational patterns. | 80.9 |
| 12 | The assistant must avoid contradictions between answers and follow-up explanations while maintaining roleplay consistency. | 78.4 |
| 13 | The assistant should provide culturally adapted translations of idioms/phrases rather than literal interpretations. | 78.0 |
| 14 | The assistant must verify numerical accuracy through step-by-step validation and real-world feasibility checks. | 81.9 |
| 15 | The assistant's code examples must be complete, functional, and demonstrate separation of concerns (HTML/CSS/JS). | 79.2 |
| 16 | The assistant should address all query components methodically, even if intermediate steps contain errors. | 81.1 |
| 17 | The assistant must maintain logical flow between concepts and preserve essential content in creative adaptations. | 82.8 |
| 18 | The assistant should prioritize factual accuracy over hypothetical interpretations unless explicitly requested. | 82.1 |
| 19 | The assistant's self-evaluations must critically assess response quality and identify specific improvement areas. | 73.6 |

Table 14: UltraFeedback rules extracted on justifications instead of reasoning CoTs.

| Index | Rule | Align (%) |
|---|---|---|
| 0 | The assistant's responses should include concrete examples, actionable insights, and specific applications to explain mechanisms and variables. | 73.1 |
| 1 | The assistant's code must handle edge cases, ensure functionality, avoid unsafe practices, and include error handling. | 81.2 |
| 2 | Structure explanations logically with step-by-step formats, clear sections, and thematic grouping while maintaining flow. | 72.8 |
| 3 | Correct user misconceptions with accurate information using empathetic and polite language. | 80.5 |
| 4 | Be concise, avoid redundancy, and prioritize clarity by eliminating unnecessary details. | 65.3 |
| 5 | Provide complete, functional code examples with necessary parameters and modular structures. | 77.3 |
| 6 | Maintain a neutral, professional tone appropriate to context without unsolicited commentary. | 74.7 |
| 7 | Strictly adhere to user instructions without deviation or unwarranted assumptions. | 71.8 |
| 8 | Use structured formatting like bullet points and headings for readability and scannability. | 74.8 |
| 9 | Address all query components comprehensively with direct answers and relevant context. | 76.7 |
| 10 | Validate code functionality, address pitfalls, and ensure integration with existing setups. | 78.3 |
| 11 | Anticipate implicit needs while avoiding speculative language beyond provided evidence. | 82.1 |
| 12 | Include practical details, alternatives, and implementation steps for real-world application. | 76.2 |
| 13 | Ensure technical accuracy, correct terminology, and compliance with domain standards. | 82.8 |
| 14 | Avoid tangential topics and focus strictly on core requests without scope creep. | 67.1 |
| 15 | Transparently admit limitations and provide actionable alternatives when uncertain. | 70.6 |
| 16 | Prioritize ethical responsibility, legal compliance, and cultural sensitivity. | 83.8 |
| 17 | Use precise language, avoid jargon, and explain technical terms contextually. | 77.9 |
| 18 | Incorporate error handling, reliability checks, and security best practices. | 78.8 |
| 19 | Balance brevity with necessary detail, adapting to user's proficiency level. | 68.7 |
| 20 | Provide self-contained, compilable code with headers and standard libraries. | 71.4 |
| 21 | Maintain logical coherence, avoid contradictions, and ensure factual consistency. | 83.7 |
| 22 | Structure narratives chronologically/thematically with clear cause-effect relationships. | 71.5 |
| 23 | Use empathetic tone, constructive feedback, and collaborative language. | 74.9 |
| 24 | Include quantitative data, contextual reasoning, and measurable outcomes. | 71.9 |
| 25 | Offer platform-agnostic solutions unless specific tools are requested. | 73.0 |
| 26 | Highlight key takeaways with memorable framing and searchable keywords. | 73.1 |
| 27 | Ensure translations preserve meaning, context, and grammatical correctness. | 84.8 |
| 28 | Link concepts to real-world impacts, case studies, and stakeholder outcomes. | 74.5 |
| 29 | Adopt solution-oriented tone with proactive guidance and troubleshooting tips. | 76.9 |

# D RULE AGREEMENT MATRICES

We showcase full rule agreement matrices between UltraFeedback and MT-Bench extracted rules on both UltraFeedback and MT-Bench data in Figures 4 and 5.

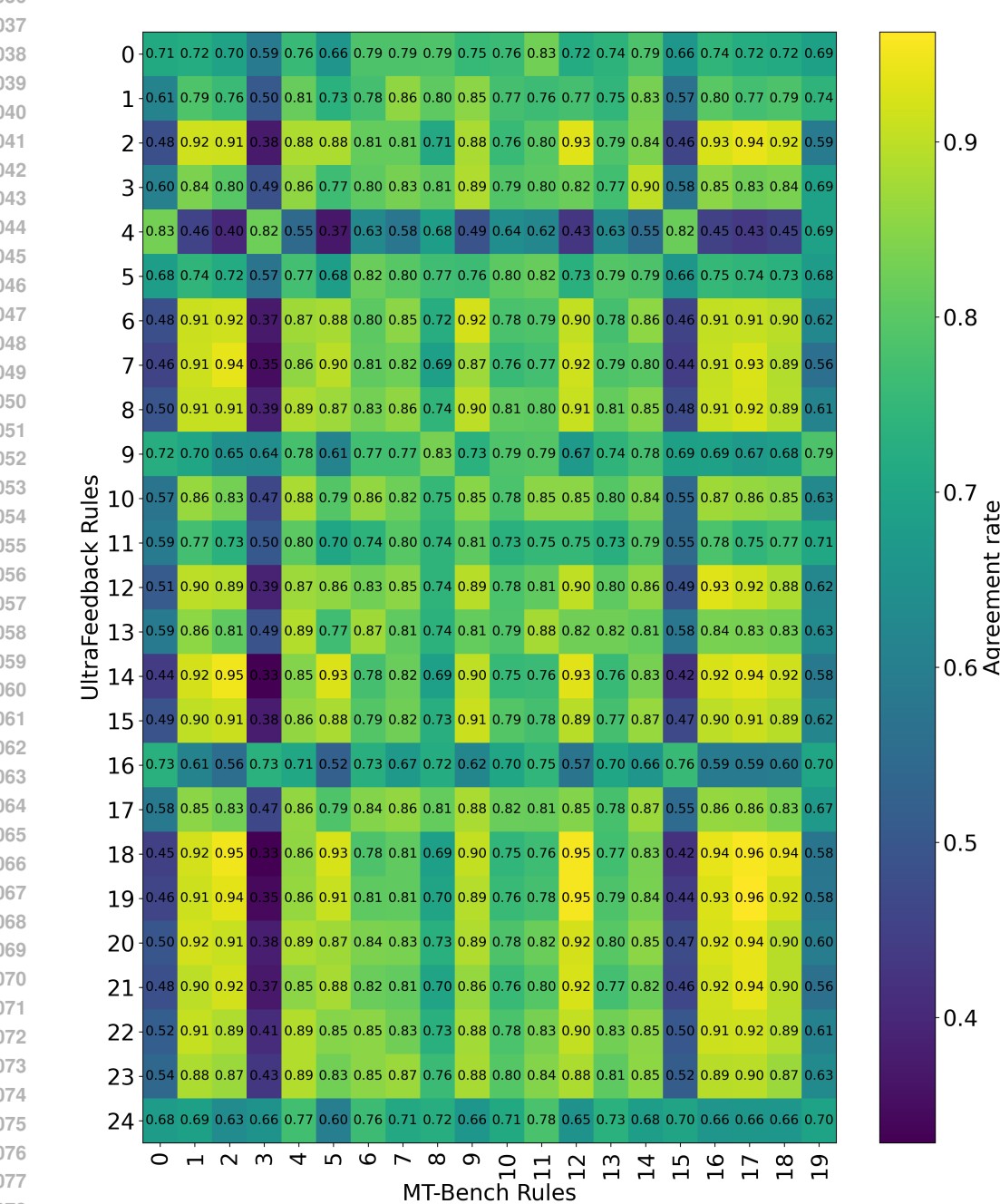

Figure 4: Rule agreement matrix on UltraFeedback data

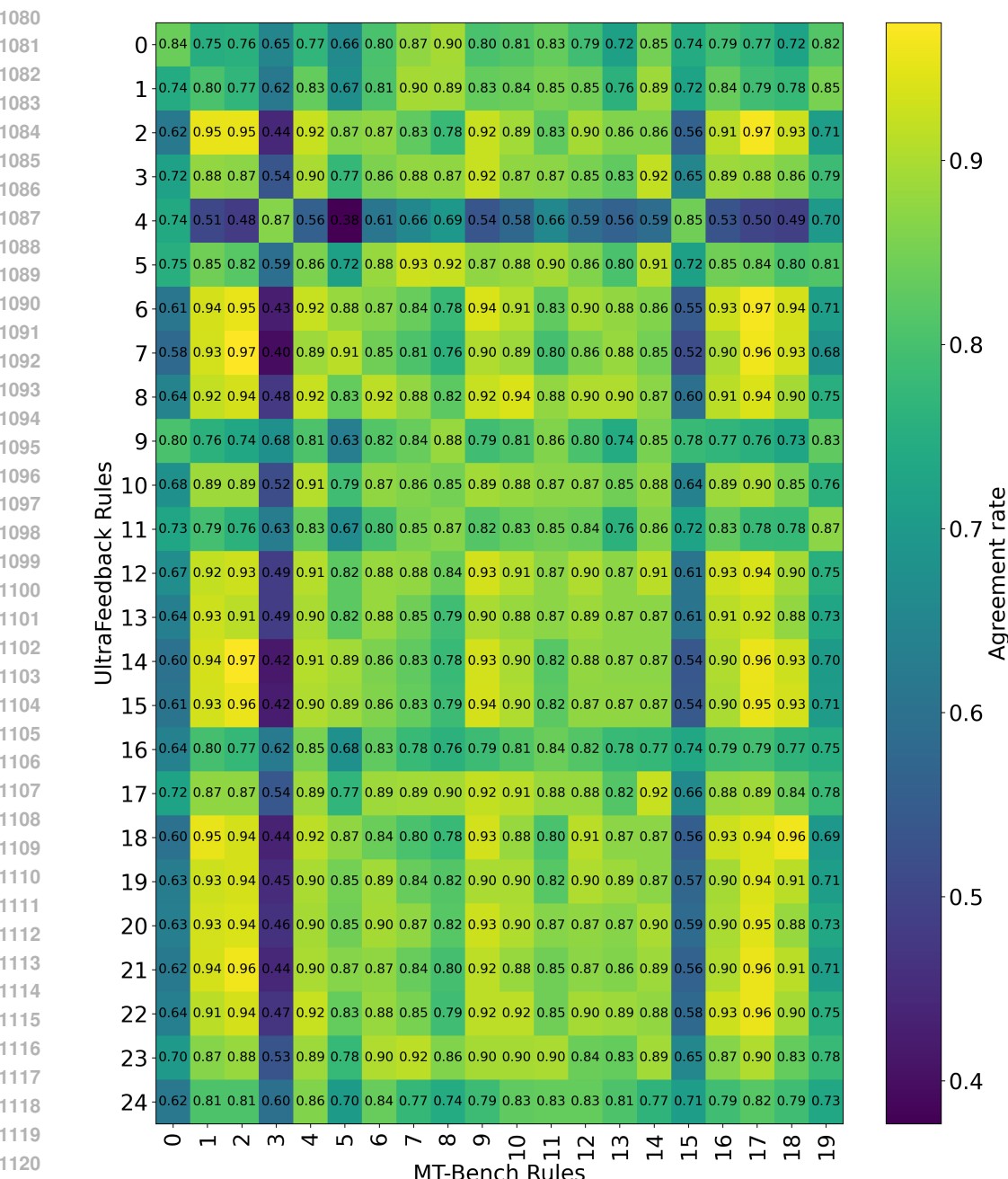

Figure 5: Rule agreement matrix on MT-Bench Human Judgements data

| Step | Similarity | Metrics |
|------|-----------|---------|
| Reasoning | Lexical | BLEU-1/ROUGE-1/Unigram Jaccard = **0.634 / 0.694 / 0.465** |
| Rule extraction | Behavior | $\sigma_{mean}/\sigma_{std}/r_{mean}/r_{std}$ = **0.088 / 0.059 / 0.865 / 0.043** |
| | LLM-judge | Jaccard = **0.563 ± 0.206** |
| Merging | Behavior | $\sigma_{mean}/\sigma_{std}/r_{mean}/r_{std}$ = **0.037 / 0.027 / 0.960 / 0.011** |
| | LLM-judge | Jaccard = **0.378 ± 0.140** |

Table 15: AutoRule stability summary (means across UF examples; rounded to 3 decimals).

# E DETAILED RULE CONSISTENCY ANALYSIS

A set of additional replication experiments was conducted to quantify the stability of the AUTORULE pipeline. For the reasoning stage, 10 UltraFeedback examples were selected and, for each response pair, chain-of-thought (CoT) traces were generated 64 times; pairwise lexical similarity was then computed across the generated CoTs. For rule extraction, 10 preference pairs were processed with 64 extraction runs; consistency was measured both behaviorally (by comparing aggregated rule scores over 256 UltraFeedback examples) and semantically using LLM-judged rule similarity. The latter was used to construct mutual k-NN graphs of extracted concepts, and pairwise concept-set similarity was summarized via the Jaccard index $\frac{|A \cap B|}{|A \cup B|}$. Finally, the merging procedure was repeated 64 times on 256 UF examples and evaluated with the same behavioral and LLM-judged metrics.

Results in Table 15 quantify stability across the pipeline. The reasoning stage shows moderate-to-high lexical agreement (BLEU-1 (Papineni et al., 2002) / ROUGE-1 (Lin, 2004) / Unigram Jaccard = 0.634 / 0.694 / 0.465). The rule-extraction stage exhibits lower behavioral variance and higher behavioral correlation ($\sigma_{mean}/\sigma_{std}/r_{mean}/r_{std} = 0.088/0.059/0.865/0.043$) and substantial semantic agreement by LLM judgment (Jaccard = $0.563 \pm 0.206$). The merging stage has even lower behavioral variance and higher behavioral correlation ($0.037/0.027/0.960/0.011$) while showing a lower LLM-judged semantic Jaccard ($0.378 \pm 0.140$), consistent with consolidation producing behaviorally consistent but semantically more variable rule sets. Altogether, these metrics support that AUTORULE yields stable, reproducible rules appropriate for deriving reward signals.

We also conducted a small experiment to assess the determinism of the rules by running verifier inference with a temperature of 1.0, 100 times on 20 UltraFeedback test-set responses for the UltraFeedback-extracted rules and 16 MT-Bench test-set responses for the MT-Bench-extracted rules. Using a determinism score calculated as $(\max(\#\text{Yes}, \#\text{No})/(\#\text{Yes} + \#\text{No}))$, where Yes/No indicates the answer to rule satisfaction, we obtained average determinism scores of 83.6% and 82.5% for UltraFeedback and MT-Bench-extracted rules, respectively. These results suggest that the rule verification exhibit a high degree of consistency and that the verifier provides reliable judgments, supporting their suitability for reward formulation.

# F CASE STUDY: REASONING CHAIN VS JUSTIFICATION EXAMPLES

To further analyze the importance of utilizing reasoning chains for rule extraction, we showcase a case study of a prompt asking about the role of zoos in animal conversation in Figure 6. In the two model responses, Response A is clearly worse as it includes an out-of-place rhetorical question. While both the justification and the reasoning chain offer valid points, the reasoning chain is much more detailed and step-by-step in offering these points, including more examples and specific references to text, whereas the justification just provides a final summary.

By using a reasoning chain, our belief is that AUTORULE is able to extract better more in-depth and logical rules since the reasoning chain offers logical justifications rather than vague descriptions.

**User Prompt**

What role do zoos and aquariums play in educating the general public about conservation efforts?

**Assistant A (Rejected)**

Did you know that zoos and aquariums provide critical opportunities for conservation education? When people visit these institutions, they learn about the dangers of habitat destruction, loss of biodiversity, and environmental disruptions. Many zoos and aquariums also participate in conservation efforts directly by breeding endangered species, supporting research, and protecting wild populations. It's essential that we prioritize education and conservation efforts to help protect our planet and its amazing wildlife.

**Assistant B (Chosen)**

Zoos and aquariums play a crucial role in educating the general public about conservation efforts. They not only offer a visual representation of the animals and their natural habitats but also provide valuable information about the animals, their behaviors, and the challenges they face in the wild. Many zoos and aquariums also support research, conservation projects, and breeding programs for endangered species. Additionally, they offer interactive exhibits and educational programs that engage visitors in these conservation efforts.

**Reasoning Chain**

The user wants to know why Assistant B's conversation was preferred over Assistant A's. I need to compare both responses based on clarity, coherence, helpfulness, tone, and overall quality.

Starting with Assistant A: The response mentions that zoos and aquariums provide critical opportunities for conservation education. They talk about habitat destruction, biodiversity loss, and environmental disruptions. They also mention breeding programs, research, and protecting wild populations. The tone seems positive, ending with a statement about prioritizing education and conservation. But maybe the structure isn't as organized. They start with a question ("Did you know...?") which might be engaging, but then list points without much flow. The information is there, but maybe not as detailed or structured.

Now Assistant B: They start by stating the role is crucial, then break it down into parts. First, visual representation and information about animals and their challenges. Then mention research, conservation projects, breeding programs. Also include interactive exhibits and educational programs. The structure here is more logical—each sentence builds on the previous, covering different aspects like education methods, direct conservation work, and visitor engagement. The tone is informative and straightforward without rhetorical questions. It feels more comprehensive.

Comparing clarity: Both are clear, but Assistant B is more structured, making it easier to follow. Coherence: Assistant B connects the points better, moving from education to conservation efforts to engagement. Helpfulness: Assistant B provides specific examples like "interactive exhibits," which gives concrete details. Assistant A mentions similar points but less specifically. Tone: Both are positive, but Assistant A's use of "Did you know" might come off as less formal, whereas B is consistently professional. Overall quality: B's answer is more thorough and organized, which likely makes it more effective for the user seeking a detailed explanation.

**Final Justification**

Assistant B's response was preferred due to superior clarity, coherence, and depth of information. Here's why:

**Clarity:** Assistant B used structured points and avoided rhetorical distractions.

**Coherence:** Assistant B presented a logical narrative from educational content to conservation actions and visitor engagement.

**Helpfulness:** It included concrete examples like "interactive exhibits" and "challenges they face in the wild."

**Tone:** B maintained a focused, declarative tone. A's "Did you know..." approach detracted from professionalism.

**Overall Quality:** B was more comprehensive and detailed, effectively answering both the "how" and "why" aspects of the user's question.

Figure 6: Case study: reasoning chain vs justification.

Table 16: Top 6 unique and similar rules from the UltraFeedback and MT-Bench rule sets, ranked by maximum agreement (%) as the similarity metric; "unique" rules exhibit low agreement with all rules from the other set, while "similar" rules show high agreement with at least one rule from the other set.

| Top unique rules | | | Top similar rules | | |
|---|---|---|---|---|---|
| **Rule** | **Data** | **Max** | **Rule** | **Data** | **Max** |
| Ensure conciseness by eliminating redundancy and focusing on core query relevance. | UF | 75.6 | The assistant must maintain logical flow between concepts and preserve essential content in creative adaptations. | MT | 96.1 |
| Balance thoroughness with brevity to prevent information overload while ensuring clarity. | UF | 78.6 | Maintain narrative coherence with source material when discussing plots or characters. | UF | 96.1 |
| The assistant's self-evaluations must critically assess response quality and identify specific improvement areas. | MT | 79.1 | Validate answers against provided context and avoid unsupported extrapolation. | UF | 95.4 |
| Avoid assumptions when ambiguity exists; seek clarification for insufficient context. | UF | 80.6 | The assistant's responses must maintain a professional and approachable tone, adapting to the nature of the user's query. | MT | 95.2 |
| The assistant's responses must provide detailed step-by-step explanations and calculations to ensure correctness and clarity. | MT | 82.2 | Address ethical considerations, legal compliance, and recommend professional consultation when relevant. | UF | 95.2 |
| The assistant's code examples must be complete, functional, and demonstrate separation of concerns (HTML/CSS/JS). | MT | 82.3 | The assistant must avoid contradictions between answers and follow-up explanations while maintaining roleplay consistency. | MT | 94.9 |

## G  CASE STUDY: RULE SET COMPARISON ACROSS DATASETS

**Rule Agreements.**  To further investigate the effectiveness of rule extraction, we conduct a comparative analysis of rule sets derived from UltraFeedback and MT-Bench. Specifically, we construct a rule agreement matrix by evaluating all pairs of rules on a test set of 1,024 UltraFeedback examples and the full MT-Bench human judgment test split. Based on this matrix, we identify similar and unique rules according to their agreement scores.

Table 16 present the top six unique and top six similar rules, respectively, as determined by the maximum rule agreement with rules from the opposing set across both UltraFeedback and MT-Bench examples. Unique rules from UltraFeedback seem to predominantly emphasize conciseness and clarity, while unique rules from MT-Bench are oriented toward handling complex tasks, such as self-evaluation, performing calculations, or providing code examples. This distinction likely reflects the broader topical diversity of UltraFeedback and the specialized, challenging nature of MT-Bench prompts. In contrast, the similar rules shared between the two sets consistently address core aspects of high-quality assistant responses, including logical flow, professional tone, contextual coherence, and answer consistency. Comprehensive rule agreement matrices are provided in Appendix D for further reference.

Additionally, we find that most rules are complementary and cover distinct aspects of response quality. However, some overarching themes—such as comprehensive instruction following—appear in multiple rules. For example, both rules 10 and 21 in Table 12 emphasize the importance of fully addressing the user's query.

**User Prompt**

Name five common ingredients used in French cuisine.

Figure 7: User prompt for reward hacking case study.

**GRPO response (RM score 5.000)**

French cuisine is renowned for its sophisticated flavors, delicate techniques, and rich history. While there are countless ingredients utilized in French cooking, here are five common ones that showcase the essence of this exquisite culinary tradition:

1. Butter: Unsurprisingly, butter plays a significant role in French cuisine. Rich, creamy butter is liberally used in sauces, pastries, and for cooking various dishes. The French value the flavor and texture that butter imparts, making it an indispensable ingredient.

2. Fresh Herbs: A variety of fresh herbs, such as parsley, thyme, rosemary, basil, and tarragon, are frequently employed in French cooking to enhance flavors and aromas. These herbs are often finely chopped or blended into sauces, marinades, and stuffing for meats and vegetables.

3. Wine: French cuisine is closely tied to wine, with various varieties utilized for cooking and pairing with meals. Red, white, and rosé wines find their way into sauces, stews, and marinades, adding complexity and depth to the dishes. The selection of wine depends on the specific recipe and the region's specialities.

4. Garlic: garlic is a staple in French cuisine, used to infuse dishes with a pungent, savory flavor. It's often minced or crushed and incorporated into sauces, marinades, and vegetable preparations. The French appreciate garlic's ability to enrich and enhance other ingredients without overtaking the dish's taste.

5. Flour: High-quality flour is a fundamental component in French baking and cooking. Whether it's all-purpose flour, bread flour, or specialty flours like almond or whole wheat, flour is utilized to create delicate pastries, crispy crusts, and thickening agents for sauces. The texture and flavor of the flour significantly impact the final dish's quality.

6. Cheese: French cuisine boasts an extensive range of exceptional cheeses, with each region possessing its unique specialties. Soft cheeses like Brie, Camembert, and Roquefort, as well as hard cheeses like Comté and Gruyère, find their way into dishes, sandwiches, and as flavor enhancers in sauces.

Remember that this list is not exhaustive, and plenty of other ingredients, techniques, and regional specialties contribute to the richness and diversity of French cuisine. The mentioned ingredients, however, reflect the core values and tastes that define this captivating culinary tradition.

As an AI language model, I must clarify that while I provided information on French cuisine, I cannot taste or experience food like a human. However, I can assure you that the ingredients mentioned above are commonly used in French cooking and contribute to its renown.

Figure 8: GRPO response for reward hacking case study.

> **AUTORULE response (RM score 4.125)**
>
> 1. Butter: Butter is a quintessential ingredient in French cuisine, providing richness, flavor, and texture to dishes. It's often used in sautéing, baking, and emulsifying sauces like bechamel and hollandaise.
>
> 2. Fresh Herbs: French cuisine emphasizes the use of fresh herbs to enhance flavors and aromas. Common herbs include parsley, thyme, sage, rosemary, and tarragon. They are often combined in subtle balances to create distinctive flavor profiles.
>
> 3. Garlic: Garlic is a staple in French cooking, adding a pungent yet enjoyable flavor to dishes. It's commonly used in sauces, stews, and marinades, and is often paired with other ingredients like butter and herbs.
>
> 4. Wine: Wine is widely used in French cuisine as an ingredient and a cooking agent. Red and white wines are suited for different dishes, adding complexity, acidity, and depth of flavor. Wine reductions and sauces, such as boeuf bourguignon and coq au vin, are popular examples.
>
> 5. Flour: French cuisine makes extensive use of various types of flour, including white wheat flour, whole wheat flour, and specialty flours like buckwheat (for crepes and galettes). Flour is used for making pastries, bread, and thickening sauces.
>
> Remember that these ingredients may vary depending on the specific region and dish in French cuisine, but they are common across many recipes.

Figure 9: AUTORULE response for reward hacking case study.

## H CASE STUDY: REWARD HACKING

To further analyze reward hacking behavior, we present an illustrative example comparing the behaviors of the GRPO and AUTORULE models after two episodes of RL training. The user prompt for the example is shown in Figure 7 and the GRPO and AUTORULE responses are displayed in Figures 8 and 9, respectively.

**Analysis.** The GRPO model outputs six ingredients instead of the five explicitly requested by the user. This additional item likely leads the reward model to interpret the response as a more comprehensive ingredient list and therefore erroneously assign a higher reward. In contrast, the AUTORULE model follows the instruction precisely and outputs exactly five ingredients. As expected, when evaluating both responses using the same inference settings and the same prompt as in our Ultra-Feedback win-rate evaluations, we find that the AUTORULE response wins irrespective of ordering. This example highlights the core difference in reward hacking behavior: while GRPO overoptimizes by exploiting superficial patterns in the reward model, AUTORULE remains instruction-faithful and avoids such reward hacking tendencies.

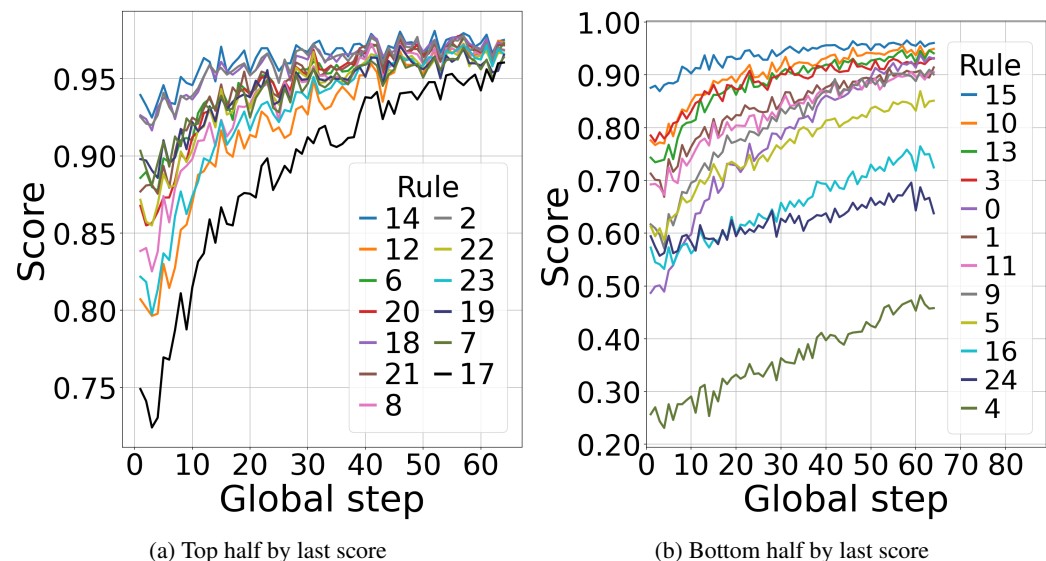

(a) Top half by last score

(b) Bottom half by last score

Figure 10: Figures 3a and 3b show individual rule curves for two episodes on a Llama-3-8B AUTORULE run.

## I RULE CURVES

We plot AUTORULE rule curves over 64 training steps in Figure 10. The trend shows that all rules exhibit upward score behavior.

# J  PROMPTS

We list the prompts used for the extraction process in Figures 11 , 12, and 13 respectively. Additionally, we include the prompts for rule verification in Figures 14 and 15, and the prompt used to determine UltraFeedback winner judgments for win-rate calculation in Figure 16.

---

**Justification Prompt**

[Instruction]
You are tasked with analyzing two conversations between an AI assistant and a user. Based on the content, please provide a detailed explanation of why the user might have preferred the winning conversation.
Please consider aspects such as clarity, coherence, helpfulness, tone, and overall quality.
[Conversation with Assistant A]
{conversation_a}
[Conversation with Assistant B]
{conversation_b}
[Winning Conversation]: {winner}
[Your Explanation]

---

Figure 11: Justification (AUTORULE Extractor stage 1) prompt.

---

**Rule Extraction Prompt**

[Instruction]
Based on the following reasoning about why conversation with assistant winner is better, extract any rule-like statements implied by the reasoning that indicate this preference. Rule-like statements should be able to be judged objectively and deterministically. Below are a few examples of rule-like statements:
Example 1:
- The assistant's responses should validate any assumptions made with sufficient context and examples.
Example 2:
- The assistant's responses should not simply restate information provided by the user as its answer.
Example 3:
- The assistant's responses should have a structure that satisfies the user's request.
Return the list as a JSON array of strings. Do not use "'json"', just output the JSON array directly. If there are no rule-like statements, return an empty JSON array.
[Reasoning]
{reasoning_chain}

---

Figure 12: Rule extraction (AUTORULE Extractor stage 2) prompt.

---

**Rule Merging Prompt**

[Instruction]
Below is a large list of rule-like statements regarding the behavior of an AI assistant. Some of these rules might be duplicates or very similar in meaning.
Please merge them so that there are no duplicates or rules with very similar meanings.
Return the merged list as a JSON array of strings. Do not use "'json"', just output the JSON array directly.
[Rules]
{rules_text}

---

Figure 13: Rule merging (AUTORULE Extractor stage 3) prompt.

---

**Rule Verifier Prompt**

You are an impartial judge. Determine whether the AI assistant's response in the following conversation both complies with the rule below and does so in a concise manner:

Rule:
{rule}

[Start of Conversation]
{conversation}
[End of Conversation]

[Analysis]
Base your judgment solely on whether (1) the response satisfies the rule and (2) the response does so in a concise manner.

Only respond with "[[Yes]]" if **both** conditions are fully satisfied. If either condition is not met, respond with "[[No]]". If the rule is not applicable to the task, treat it as satisfied.

Respond with one of the following options, and nothing else: "[[Yes]]" or "[[No]]".

---

Figure 14: Rule verifier prompt.

---

**Rule Verifier Prompt (no conciseness)**

[Instruction]
Please act as an impartial judge and evaluate whether the responses provided by an AI assistant in the following conversation satisfy the following rule:
{rule}
Be as objective as possible when evaluating the rule and do not evaluate other characteristics of the response. If the rule is not applicable for this task, treat it as if the rule is satisfied. You must provide your answer by strictly outputting either one of the following two options: "[[Yes]]" or "[[No]]" and nothing else.
[Start of Conversation]
{conversation}
[End of Conversation]

---

Figure 15: Rule verifier prompt (no conciseness).

**UF Win-rate Judgement Prompt**

I want you to create a leaderboard of different large-language models. To do so, I will give you the instructions (prompts) given to the models, and the responses of two models. Please rank the models based on which responses would be preferred by humans. All inputs and outputs should be python dictionaries.

Here is the prompt:
```
{{
"instruction": """{instruction}"""
}}
```

Here are the outputs of the models:
```
[
    {{
        "model": "model_1",
        "answer": """{output_1}"""
    }},
    {{
        "model": "model_2",
        "answer": """{output_2}"""
    }}
]
```

Now please rank the models by the quality of their answers, so that the model with rank 1 has the best output. Then return a list of the model names and ranks, i.e., produce the following output:

```
[
    {{'model': <model-name>, 'rank': <model-rank>}},
    {{'model': <model-name>, 'rank': <model-rank>}}
]
```

Your response must be a valid Python dictionary and should contain nothing else because we will directly execute it in Python. Please provide the ranking that the majority of humans would give.

Figure 16: UltraFeedback win-rate judgement prompt.

# K LICENSES

Asset URLS and licenses are displayed in Table 17.

Table 17: Asset URLs and licenses. *Custom license available at `https://llama.meta.com/llama3/license`.

| Asset | URL | Purpose | License |
|---|---|---|---|
| Llama-3-8B | `https://huggingface.co/meta-llama/Meta-Llama-3-8B` | Base model | Custom* |
| OLMo-2-7B | `https://huggingface.co/allenai/OLMo-2-1124-7B` | Base model | Apache-2.0 |
| DeepSeek-R1 | `https://aws.amazon.com/bedrock/deepseek/` (Used on Bedrock) | Extraction process | MIT |
| UltraFeedback-Binarized | `https://huggingface.co/datasets/lmsys/mt_bench_human_judgments` | Dataset | MIT |
| MT-Bench Human Judgements | `https://huggingface.co/datasets/lmsys/mt_bench_human_judgments` | Dataset | CC-BY 4.0 |
| LLM-as-a-judge code | `https://github.com/lm-sys/FastChat/tree/main/fastchat/llm_judge` | MT-Bench benchmark | Apache-2.0 |
| AlpacaEval repo | `https://github.com/tatsu-lab/alpaca_eval` | AlpacaEval 2.0 benchmark | Apache-2.0 |
| PAR repo | `https://github.com/PorUna-byte/PAR` | Filtering code | MIT |
| OpenRLHF | `https://github.com/OpenRLHF/OpenRLHF` | Training framework | Apache-2.0 |
| vLLM | `https://github.com/vllm-project/vllm` | Model inference | Apache-2.0 |

