# OpenReview forum: "AutoRule: Reasoning Chain-of-Thought Extracted Rule-based Rewards Improve Preference Learning"
_ICLR.cc/2026/Conference — Submitted to ICLR 2026_

### Official Review · Reviewer_SJ6k · 2025-10-27

**Soundness:** 3
**Presentation:** 2
**Contribution:** 3
**Rating:** 4
**Confidence:** 3

**Summary:**

This paper presents a pipeline for extracting rule-based rewards for preference alignment of LLMs. The pipeline employs a reasoning LLM (e.g. DeepSeek-R1) to generate reasoning chains for a subset of training samples, extract rules from the reasoning chains, and then merge the extracted rules into a concrete list, which is used by a verifier LLM to produce 0/1 rewards during training. The paper examines the effectiveness of this pipeline on UltraFeedback, MT-Bench, and AlpacaEval2.0 and presents a series of in-depth analysis for the quality of rules, consistency of rules, and pipeline ablation.

**Strengths:**

1. The idea of extracting reward rules from reasoning models is an interesting idea, since rules are compact and likely to be generalizable.

2. The proposed pipeline is straight-forward and concise.

**Weaknesses:**

1. The presentation needs improvement. For example, even though the author mention RaR and RLCF, two concurrent word dedicated on automatic rule construction, from line 145-146 it is unclear how the proposed method differs from them. In addition, the inclusion of conciseness as a reference is not mentioned in section 3, but it is experimentally analyzed in line 375. There are also some minor mistakes. The symbol $y$ is defined to be output sequences, so I think in line 166 and 176 the correct notation is $(o,r)\sim\pi_\phi(\cdot|x)$. The 1 and 2 shown in line 174 and line 175 are also confusing.

2. Although this method is claimed to be resolving reward hacking, the only evidence provided are the learning curves presented in Fig 3a & 3b, where the model reward of alternative methods begin to decline after training proceed but the model reward of the proposed method keep improving. Although I agree that such results are positive, their connection with reward hacking needs further illustration.

3. The proposed pipeline has some space to improve. See questions below.

**Questions:**

1. The results shown in Table 2 suggest that the proposed method is label efficient, if we regard token from the teacher model as labels. But how do you determine the amount subset of data to be used for rule extraction? Is the size of such subset correlated with the overall performance?

2. Rather than random selection, is there a better way to select samples in this subset?

3. Section 5.3 presents quantitative evaluation for the extracted rules. What do you mean by "from different rule lists" in line 431?

4. I think generalizability is a more intuitive name for the equation in line 424. Could you provide results similar to fig 2a, but in the setting of cross-domain generalization? In other words, evaluate rules extracted from UltraFeedback on other datasets.

---

> ### Author Response · Authors · 2025-11-24
>
> Thank you for your thorough review and constructive feedback. We appreciate your positive remarks on the novelty of extracting reward rules from reasoning models, as well as the straightforward and concise nature of our proposed pipeline. Please find our detailed responses to your comments and questions below.
>
> **Weaknesses:**
>
> 1. **(W1)** Thank you for pointing out the presentation issues. We have corrected the notation mistakes and clarified confusing numbers (e.g., the “1" and “2” which represents the relative positioning of the chosen response with respect to the rejected response). We also revised Section 3 to explicitly introduce the conciseness reference used in the verifier. Regarding the distinction from RaR and RLCF: unlike these methods, which generate rules per prompt, our approach uses reasoning chains to extract rules once and then merges them into a unified global rule set. This makes our method significantly more teacher-model efficient, as demonstrated in Table 2\. We have updated the paper to make these methodological differences clearer.
> 2. (**W2)** Following prior work, a standard way to detect reward hacking is to track a true (gold) metric over the course of training and observe whether it initially increases and then later declines—a characteristic pattern reported in the PAR \[1\] and InfoRM \[2\] papers. In our experiments, we observe that baseline methods indeed exhibit this downward trend in gold metrics, while AutoRule continues to improve. Figures 3c and 3d **effectively illustrate that the baseline is “hacking” the reward by exploiting spurious local optima where the win rate decreases despite the reward increasing.** We have clarified this connection in the revised explanation.
> 3. **(W3)** We address your questions below.
>
> **Questions:**
>
> 1. **(Q1)** In Figure 2c, we evaluate the average behavioral agreement between rule sets extracted from different training subset sizes and the rule set derived from 256 examples. We find consistently high agreement across all scales, indicating that the rule-extraction process is robust to the number of training examples used. In our experiments, we selected 256 examples as a setting for demonstrating the effectiveness of the method.
> 2. **(Q2)** Beyond random selection, more principled strategies could be explored. One possibility is influence-based selection, where we choose examples predicted to have the largest impact on downstream reward-model accuracy or policy performance. Another direction is diversity-based selection, where we select examples that broadly cover the task distribution, for instance by clustering samples in embedding space and choosing representatives from each cluster. We view these as promising extensions for future work.
> 3. **(Q3)** By “different rule lists,” we refer to comparing two separate rule sets for consistency. For extraction-scale analyses, this means comparing the rule set extracted from n training examples with the rule set extracted from 256 examples. For the correlation metrics in our stability evaluation, we compute consistency pairwise across the five independent runs, yielding a cross-run agreement measure. We have clarified this wording in the paper.
>
> 4. **(Q4)** Figure 2a has been updated to present cross-domain alignment results in addition to the in-domain measurements. The observed alignment indicates that the extracted rules generalize effectively across datasets, demonstrating that they capture accurate and transferable decision principles. We also observe higher alignment overall when evaluating against MT preferences, likely due to the expert-level annotations used in that dataset.
>
> References:
>
> 1. Jiayi Fu, Xuandong Zhao, Chengyuan Yao, Heng Wang, Qi Han, and Yanghua Xiao. Reward shaping to mitigate reward hacking in rlhf, 2025\. URL [https://arxiv.org/abs/2502.18770](https://arxiv.org/abs/2502.18770).
> 2. Yuchun Miao, Sen Zhang, Liang Ding, Rong Bao, Lefei Zhang, and Dacheng Tao. Inform: Mitigating reward hacking in rlhf via information-theoretic reward modeling. In A. Globerson, L. Mackey, D. Belgrave, A. Fan, U. Paquet, J. Tomczak, and C. Zhang (eds.), *Advances in Neural Information Processing Systems*, volume 37, pp. 134387–134429. Curran Associates, Inc., 2024\. URL [https://proceedings.neurips.cc/paper\_files/paper/2024/file/f25d75fc760aec0a6174f9f5d9da59b8-Paper-Conference.pdf](https://proceedings.neurips.cc/paper_files/paper/2024/file/f25d75fc760aec0a6174f9f5d9da59b8-Paper-Conference.pdf).

---

> > ### Author Response · Authors · 2025-11-29
> >
> > As an addendum to our discussion of W2, to further analyze reward hacking behavior, we present an illustrative example comparing the behaviors of the GRPO and AutoRule models after two episodes of RL training. We included this case study in our appendix.
> >
> > **User Prompt:** Name five common ingredients used in French cuisine.
> >
> > **GRPO response (RM score 5.000):**
> >
> > French cuisine is renowned for its sophisticated flavors, delicate techniques, and rich history. While there are countless ingredients utilized in French cooking, here are five common ones that showcase the essence of this exquisite culinary tradition:
> >
> > 1\. Butter: Unsurprisingly, butter plays a significant role in French cuisine. Rich, creamy butter is liberally used in sauces, pastries, and for cooking various dishes. The French value the flavor and texture that butter imparts, making it an indispensable ingredient.
> >
> > 2\. Fresh Herbs: A variety of fresh herbs, such as parsley, thyme, rosemary, basil, and tarragon, are frequently employed in French cooking to enhance flavors and aromas. These herbs are often finely chopped or blended into sauces, marinades, and stuffing for meats and vegetables.
> >
> > 3\. Wine: French cuisine is closely tied to wine, with various varieties utilized for cooking and pairing with meals. Red, white, and rosé wines find their way into sauces, stews, and marinades, adding complexity and depth to the dishes. The selection of wine depends on the specific recipe and the region's specialities.
> >
> > 4\. Garlic: garlic is a staple in French cuisine, used to infuse dishes with a pungent, savory flavor. It's often minced or crushed and incorporated into sauces, marinades, and vegetable preparations. The French appreciate garlic's ability to enrich and enhance other ingredients without overtaking the dish's taste.
> >
> > 5\. Flour: High-quality flour is a fundamental component in French baking and cooking. Whether it's all-purpose flour, bread flour, or specialty flours like almond or whole wheat, flour is utilized to create delicate pastries, crispy crusts, and thickening agents for sauces. The texture and flavor of the flour significantly impact the final dish's quality.
> >
> > 6\. Cheese: French cuisine boasts an extensive range of exceptional cheeses, with each region possessing its unique specialties. Soft cheeses like Brie, Camembert, and Roquefort, as well as hard cheeses like Comté and Gruyère, find their way into dishes, sandwiches, and as flavor enhancers in sauces.
> >
> > Remember that this list is not exhaustive, and plenty of other ingredients, techniques, and regional specialties contribute to the richness and diversity of French cuisine. The mentioned ingredients, however, reflect the core values and tastes that define this captivating culinary tradition.
> >
> > As an AI language model, I must clarify that while I provided information on French cuisine, I cannot taste or experience food like a human. However, I can assure you that the ingredients mentioned above are commonly used in French cooking and contribute to its renown.
> >
> > **AutoRule response (RM score 4.125):**
> >
> > 1\. Butter: Butter is a quintessential ingredient in French cuisine, providing richness, flavor, and texture to dishes. It's often used in sautéing, baking, and emulsifying sauces like bechamel and hollandaise.
> >
> > 2\. Fresh Herbs: French cuisine emphasizes the use of fresh herbs to enhance flavors and aromas. Common herbs include parsley, thyme, sage, rosemary, and tarragon. They are often combined in subtle balances to create distinctive flavor profiles.
> >
> > 3\. Garlic: Garlic is a staple in French cooking, adding a pungent yet enjoyable flavor to dishes. It's commonly used in sauces, stews, and marinades, and is often paired with other ingredients like butter and herbs.
> >
> > 4\. Wine: Wine is widely used in French cuisine as an ingredient and a cooking agent. Red and white wines are suited for different dishes, adding complexity, acidity, and depth of flavor. Wine reductions and sauces, such as boeuf bourguignon and coq au vin, are popular examples.
> >
> > 5\. Flour: French cuisine makes extensive use of various types of flour, including white wheat flour, whole wheat flour, and specialty flours like buckwheat (for crepes and galettes). Flour is used for making pastries, bread, and thickening sauces.
> >
> > Remember that these ingredients may vary depending on the specific region and dish in French cuisine, but they are common across many recipes.
> >
> > (Analysis in next comment)

---

> > > ### Author Response · Authors · 2025-11-29
> > >
> > > **Analysis:** The GRPO model outputs six ingredients instead of the five explicitly requested by the user. This additional item likely leads the reward model to interpret the response as a more comprehensive ingredient list and therefore erroneously assign a higher reward. In contrast, the AutoRule model follows the instruction precisely and outputs exactly five ingredients. As expected, when evaluating both responses using the same inference settings and the same prompt as in our UltraFeedback win-rate evaluations, we find that the AutoRule response wins irrespective of ordering. This example highlights the core difference in reward hacking behavior: while GRPO overoptimizes by exploiting superficial patterns in the reward model, AutoRule remains instruction-faithful and avoids such reward hacking tendencies.

---

### Official Review · Reviewer_V4rY · 2025-10-30

**Soundness:** 2
**Presentation:** 2
**Contribution:** 2
**Rating:** 4
**Confidence:** 3

**Summary:**

This paper introduces an automated framework for extracting explicit natural language rules from human preferential data and using them as additional rewards in RLHF. By doing so, they provide some interpretable or constraint like rewards and hence mitigating reward hacking.

**Strengths:**

* The paper proposes a fully automated pipeline to extract the rules from the data. The amount of manual engineering is pretty nominal
* The extracted rules are human interpretable and seem to be aligned with known good practices for llm responses.

**Weaknesses:**

* The results are quite marginal and limited. Llama 3 8B is quite old at this point. And the AE LC Win rate is quite low, compared to Llama 3 8B Instruct.
* THe results seem to raise a question whether the advantage of rules is mainly effective in out of distribution or extreme scenarios, rather than in distribution (seems contrary as the rules are derived from this distribution).
* The conciseness constraint added to the verifier is an implicit design choice. It may bias the model toward shorter responses and is not extensively evaluated
* The cited parallel works (RaR and RLCF) show larger improvements on their respective benchmarks, comparing those methods with the presented evaluation would be needed to contextualize the contributions of this method.

**Questions:**

1. How sensitive is the rule set to the choice of teacher model?  Would using a smaller or different reasoning model significantly change the rules or performance?
2. Can one rule set extracted from UltraFeedback generalize to other domains?
3. After merging, are the final rules entirely complementary, or did you observe any cases of overlap/conflict between rules?

---

> ### Author Response · Authors · 2025-11-24
>
> Thank you for your thoughtful review and constructive feedback. We appreciate your positive comments regarding our fully automatic pipeline as well as our human-interpretable and aligned rules. Please find our detailed responses to your comments and questions below.
>
> **Weaknesses:**
>
> 1. **(W1)** While Llama-3-8B is indeed an older model, the instruct version remains a strong one among those that have not undergone extensive and complex post-training. In addition, OLMo-2-7B is a recent open release from AI2 (with its next-generation successor launched on November 20, 2025). Our results show consistent improvements on both architectures. As shown in Table 1, AutoRule yields substantial gains in LC win rate vs. GPT-4, with 20.7% and 7.5% relative improvements for Llama-3-8B and OLMo-2-7B, respectively, over their baselines. For comparison, Llama-3-8B-Instruct achieves only about a 6% relative improvement in LC win rate vs GPT4 over our method—while relying on unreleased datasets.
> 2. **(W2)** Our results in Table 1 show that the rules are effective in-distribution, as evidenced by improvements in UltraFeedback win rate over SFT for both OLMo-2-7B and Llama-3-8B. Additionally, because the rules are expressed in natural language and capture broadly applicable instruction-following principles, they are expected to generalize beyond the training distribution.
> 3. **(W3)** Conciseness is an important property to control: reward models frequently exhibit undesirable length bias, and explicitly prompting for conciseness helps correct this issue in a principled way rather than as ad-hoc tuning. Empirically, we observe clear differences—at the final training step, the model trained with conciseness produces responses averaging 241.5 tokens, versus 368.7 tokens without the constraint. We also evaluate the concise vs. non-concise variants in Table 5 (“Rule Reward Construction”), and observe consistent performance gains across benchmarks, indicating that conciseness prompting meaningfully contributes to reward quality. We have also examined conciseness in models trained solely with learned reward models, comparing a baseline model against variants using length-driven hyperparameter tuning and a length penalty, and found that AutoRule outperforms these alternatives.
> 4. **(W4)** We compare against RaR \[1\]  in Table 1, where our method achieves consistent improvements across all shared benchmarks. We also updated Table 1 to include RLCF \[2\] results without program verification, as this configuration attains the strongest performance on general conversational assistance benchmarks, which are most similar to our evaluation settings. Our method likewise outperforms RLCF.
>
> OLMo-2-7B base model:
>
> |  | UF Win Rate | AlpacaEval2.0 (vs SFT) | AlpacaEval2.0 (vs GPT 4\) | MT-Bench (40-question subset) |
> | :---- | :---: | :---: | :---: | :---: |
> |  | **vs SFT** | **LC WR (WR)** | **LC WR (WR)** | **Avg (Turn 1/Turn 2\)** |
> | **AutoRule** | **79.4** | **81.6** (83.9) | **20.0** (15.9) | **7.03** (7.25/6.80) |
> | RaR-Implicit | 66.2 | 58.3 (63.5) | 9.8 (8.8) | 6.62 (7.24/6.00) |
> | RaR-Explicit | 70.3 | 68.9 (77.9) | 13.0 (14.1) | 6.68 (7.30/6.05) |
> | RLCF\* | 71.4 | 67.9 (80.1) | 13.1 (18.5) | 7.01 (7.53/6.50) |
>
> Llama-3-8B base model:
>
> |  | UF Win Rate | AlpacaEval2.0 (vs SFT) | AlpacaEval2.0 (vs GPT 4\) | MT-Bench (40-question subset) |
> | :---- | :---: | :---: | :---: | :---: |
> |  | **vs SFT** | **LC WR (WR)** | **LC WR (WR)** | **Avg (Turn 1/Turn 2\)** |
> | **AutoRule** | **77.2** | **77.0** (83.3) | **21.6** (18.6) | **7.85** (7.88 / 7.83) |
> | RaR-Implicit | 63.2 | 59.4 (66.0) | 13.6 (11.5) | 7.11 (7.41 / 6.80) |
> | RaR-Explicit | 70.4 | 64.6 (78.0) | 13.0 (14.1) | 7.06 (7.33 / 6.80) |
> | RLCF\* | 71.3 | 66.4 (83.7) | 17.9 (25.0) | 7.34 (7.90 / 6.78) |
>
> \*Variant w/o code, strongest reported performance on conversational assistance benchmarks.
>
> (Response continued in next comment)

---

> > ### Author Response · Authors · 2025-11-24
> >
> > **Questions:**
> >
> > 1. **(Q1)** We conducted an additional experiment using Qwen3-8B for the AutoRule extraction pipeline. The resulting performance is reported below and included in the ablation section. This demonstrates that the rule-extraction process remains effective even with a smaller teacher model. We have included this result in the ablation section.
> >
> > |  | UF Win Rate | AlpacaEval2.0 (vs SFT) | AlpacaEval2.0 (vs GPT 4\) |
> > | :---- | :---: | :---: | :---: |
> > | Teacher Model | **vs SFT** | **LC WR (WR)** | **LC WR (WR)** |
> > | *Deepseek-R1* | 77.2 | **77.0** (83.3) | 21.6 (18.6) |
> > | Qwen3-8B | **78.5** | 76.1 (82.3) | **22.0** (18.6) |
> >
> > 2. **(Q2)** Yes. As shown in Table 1, a rule set extracted from UltraFeedback improves the AlpacaEval length-controlled win rate, despite AlpacaEval using a different evaluation distribution. This suggests that the rules capture broadly applicable instruction-following principles that generalize beyond the training dataset.
> >
> > 3. **(Q3)** We present the complete rule sets extracted from UltraFeedback and MT-Bench in Appendix Tables 12 and 13\. Qualitatively, most rules are complementary and cover distinct aspects of response quality. Some overarching themes—such as comprehensive instruction following—appear in multiple rules. For example, both rules 10 and 21 in Table 12 emphasize the importance of fully addressing the user’s query.
> >
> > References:
> > 1. Anisha Gunjal, Anthony Wang, Elaine Lau, Vaskar Nath, Bing Liu, and Sean Hendryx. Rubrics as rewards: Reinforcement learning beyond verifiable domains, 2025\. URL [https://arxiv.org/abs/2507.17746](https://arxiv.org/abs/2507.17746).
> > 2. Vijay Viswanathan, Yanchao Sun, Shuang Ma, Xiang Kong, Meng Cao, Graham Neubig, and Tong-shuang Wu. Checklists are better than reward models for aligning language models, 2025\. URL [https://arxiv.org/abs/2507.18624](https://arxiv.org/abs/2507.18624).

---

### Official Review · Reviewer_EXre · 2025-11-01

**Soundness:** 3
**Presentation:** 3
**Contribution:** 3
**Rating:** 6
**Confidence:** 3

**Summary:**

This paper proposes a methodology for automatically generating a set of grading rules based on a preference dataset. To do so, the preferred and dispreferred responses are presented to a reasoning model which is instructed to justify the preference. From the reasoning chain elicited, concrete rules are extracted and merged with other existing rules. These rules are then used during reward model inference. The authors find that the rules generated by AutoRule lead to higher quality reward signals than other rule extraction / reward shaping methodologies.

**Strengths:**

* Extracting preference rules over a whole preference dataset is both novel and timely given recent research on rubric based optimization.
* AutoRule leads to impressive performance gains in terms of best performance and robustness to overoptimization
* The authors conduct thorough evaluations and ablations which robustly demonstrate their claims.

**Weaknesses:**

* One potential confounder is that it's unclear how much of the performance gain comes from the utility of the autogenerated rules versus the amount of inference compute spent. Namely, autorule requires doing a forward pass per rubric item. It would be useful to have an inference cost fixed evaluation, potentially by varying the thinking length of a normal llm judge.

**Questions:**

* Did the authors experiment with how RM performance is related to the number of generated rules? I would be interested in understanding the scaling behavior there.

---

> ### Author Response · Authors · 2025-11-24
>
> Thank you for your thoughtful review and constructive feedback. We appreciate your positive assessment of the novelty of generating a unified rule set, the strong performance gains, and the thoroughness of our evaluations and ablations. Below, we address your concerns and questions in detail.
>
> **Weaknesses:**
>
> 1. **(W1)** For clarity, the inference compute is identical for AutoRule and baseline models, since they have the same number of parameters. To specifically address the judge compute during reward computation, we conducted an additional experiment designed to approximately match the compute used by AutoRule with a 5-rule rule set. Due to the 8k context-length limit of the judge model, we cannot fully match the compute corresponding to the 25-rule configuration used in the main setting by solely using thinking. Instead, we evaluate two alternatives: (1) a single LLM judge prompted with an detailed reasoning guide, inducing \~1.5K tokens of “thinking”; and (2) an ensemble of three LLM judge calls, each prompted to use a \~300-token reasoning budget, which aligns more closely with the model’s natural chain-of-thought length.
>
> |  | UF Win Rate | AlpacaEval2.0 (vs SFT) | AlpacaEval2.0 (vs GPT 4\) | Estimated FLOPs |
> | :---- | :---: | :---: | :---: | :---: |
> |  | **vs SFT** | **LC WR (WR)** | **LC WR (WR)** | **Per reward score** |
> | **AutoRule (25 rules)** | **77.2** | **77.0** (83.3) | **21.6** (18.6) | 2.00 x 10^14 |
> | AutoRule (5 rules) | 69.3 |  75.7 (76.1) | 21.0 (13.6) | 4.01 x 10^13 |
> | LLM judge (long, n \= 1\) | 61.9 | 53.6 (60.5) | 11.6 (9.9) | 4.60 x 10^13 |
> | LLM judge (short, n \= 3\) | 64.0 | 60.7 (79.0) | 16.9 (27.9) | 4.27 x 10^13 |
>
> Although AutoRule (5 rules) performs worse in win rate for UltraFeedback compared to the original AutoRule setting (25 rules), both LLM-judge baselines perform substantially worse than the 5-rule setting in both benchmarks, despite using comparable or greater inference compute. The short-thought ensemble in particular exhibits clear signs of reward hacking, demonstrating that simply increasing reasoning length or using more judge passes does not reliably improve reward quality. We have added this experiment to the ablation section.
>
> Additionally, the results in the reward hacking analysis section—where the baseline model’s UltraFeedback win rate and AlpacaEval LC WR decreases after an additional training episode—demonstrates that simply adding more compute by making additional passes over the data is ineffective. Overall, these findings suggest that AutoRule’s improvements do not stem from compute alone. Instead, the gains come from the structure and granularity of the rule-based feedback: auto-generated rules provide more consistent, systematic, and fine-grained evaluative signals than basic judge reasoning, which tends to be coarse, noisy, and more susceptible to over-optimization.
>
> **Questions:**
>
> 1. **(Q1)** We conducted a scaling experiment evaluating how rule-set size affects alignment accuracy. For rule sets of size 5, 25, and 50, we report how often the aggregated rule score assigns a higher, equal, or lower score to the preferred response on 256 UF examples:
>
> |  | | Preferred Response Score Margin  | |
> | :---: | :---: | :---: | :---: |
> | | **\> 0** | **\= 0** | **\< 0** |
> | 5 rules | 112 (43.8%) | 88 (34.4%) | 56 (21.9%) |
> | *25 rules*  | 152 (59.4%) | 30 (11.7%) | 74 (28.9%) |
> | 50 rules | 159 (62.1%) | 14 (5.5%) | 83 (32.4%) |
>
> We observe a large improvement from 5 → 25 rules, indicating that a sufficient number of rules is needed to meaningfully differentiate preferences. However, the gain from 25 → 50 rules is marginal, with diminishing returns and slightly more contradictory signals (higher Gap \< 0 count). Increasing rule count also reduces ties, as expected, but does not continue improving accuracy. We added this experiment to the rule consistency section.
>
> We also evaluated the 5-rule and 50-rule variants in downstream performance comparisons:
>
> |  | UF Win Rate | AlpacaEval2.0 (vs SFT) | AlpacaEval2.0 (vs GPT 4\) |
> | :---- | :---: | :---: | :---: |
> |  | **vs SFT** | **LC WR (WR)** | **LC WR (WR)** |
> | 5 rules | 69.3 | 75.7 (76.1) | 21.0 (13.6) |
> | **25 rules** | **77.2** | 77.0 (83.3) | 21.6 (18.6) |
> | 50 rules | 74.9 | **77.1** (83.0) | **23.5** (19.9) |
>
> These results show that AutoRule’s downstream performance is sensitive to the number of extracted rules. While AlpacaEval metrics remain relatively stable across configurations, the UF win-rate varies more noticeably. We hypothesize that the 5-rule configuration contains too few applicable rules to provide sufficiently detailed guidance, whereas the 50-rule configuration introduces many rules that are rarely applicable, diluting the reward signal. The 25-rule setting strikes a stronger balance, offering enough coverage without excessive noise. We have incorporated analysis on rule count into the ablation section.

---

### Official Review · Reviewer_zUE9 · 2025-11-01

**Soundness:** 3
**Presentation:** 2
**Contribution:** 3
**Rating:** 8
**Confidence:** 2

**Summary:**

This paper proposes an automated framework that extracts rule-based rewards from preference feedback, removing the need for manual rule design. It uses a three-stage pipeline that contains reasoning-based interpretation, rule extraction, and synthesis. The pipeline is followed by LM verification to generate auxiliary rewards. The method boosts alignment performance (+25.6% vs GPT-4 on AlpacaEval 2.0, +6.1% on MT-Bench), reduces reward hacking, and improves interpretability, with all code and models open-sourced.

**Strengths:**

The proposed method has significant gains on AlpacaEval 2.0 and MT-Bench for Llama-3-8B and Olmo-2-7B.

The proposed method has substantial implications for the community: it provides explicit, human-readable constraints that explain policy behavior.

Open-sourced rules, code, and checkpoints.

**Weaknesses:**

The authors argue it is the first fully automated rule-extraction system for RLHF/post-training. However, such pipelines are pretty industrial; therefore, the protocol and engineering efforts might not be innovative for the community.

The multi-stage design and instruction-following nature did provide logical transparency of the pipeline, but the paper did not clearly illustrate the merits of such designs. It is conceptually appealing, but I was unable to digest the evidence presented in the main body and the appendix.

**Questions:**

It is out of scope for this paper, but it would be very helpful if the dependency on the reward function were clarified.

In many real-world datasets, binary reward models, e.g., the Bradley-Terry (BT) model, are known to be subject to 'intransitivity' because they rely on scalar variables, which assume all preferences are transitive.
- The literature below studied representative preference datasets in the real world, where the 'transitive' relationship between preference annotations may not always hold.
- https://arxiv.org/abs/2409.19325 (Duan et al, 2017)

---

> ### Author Response · Authors · 2025-11-24
>
> Thank you for your thorough review and constructive feedback. We appreciate your positive feedback regarding the significant benchmark gains and the implications of our method in providing explicit human-readable constraints that explain policy behavior. Please find our detailed responses to your comments and questions below.
>
> **Weaknesses:**
>
> 1. **(W1)** While parts of the pipeline may resemble established industrial practices, our contribution is not in engineering novelty for its own sake, but in demonstrating that reasoning chains can be systematically leveraged to derive a coherent, human-readable rule set directly from preference data. The engineering challenge is not merely prompting a model to output rules, but integrating rule-based rewards into existing RLHF training frameworks. We have revised the paper to clarify this conceptual contribution and how the engineering goes beyond standard industrial heuristics.
> 2. **(W2)** We apologize for the lack of clarity regarding the motivation for our multi-stage design. A single-stage approach that asks an LLM to output the final merged rules directly can lead to unstable behavior: the model lacks a clear decomposition of subgoals and may not reliably follow the intended structure. Moreover, such an approach requires concatenating conversations across all examples into a single prompt, which increases context length significantly and could degrade LLM effectiveness. Our pipeline instead deliberately breaks the task into interpretable steps—(1) reasoning about the preference decision, (2) extracting rule-like statements from that reasoning, and (3) merging those rules across examples—to ensure that each component is well-scoped and that the LLM can focus on one objective at a time. We have updated the paper to better explain the motivation behind this design.
>
> **Questions:**
> 1. **(Q1)** Thank you for the clarification request and for pointing us to the relevant literature. Your comment highlights a key limitation of many learned reward models, which assume transitivity by representing preferences with a single latent scalar. In contrast, a key element of our approach is that AutoRule does not collapse preferences into a single score. Instead, we assume that preferences arise from a set of underlying principles, which we operationalize as “rules.” Each rule corresponds to one dimension of desirable behavior, and a human preference judgment reflects a combination or tradeoff among these principles. Under this perspective, cyclic or seemingly inconsistent preferences are not noise but rather evidence of the multi-dimensional structure underlying human evaluations, where people weigh different principles differently.

---

> > ### Comment · Reviewer_zUE9 · 2025-11-26
> >
> > I thank the authors for their candid response to my questions.
> >
> > For Q2, is this topic discussed or linked to some specific sections in the manuscript? I cannot find the respective parts in the context of the article.

---

> ### Author Response · Authors · 2025-11-26
>
> Thank you for the question. To ensure we address it precisely, could you clarify what you are referring to as Q2?

---

> > ### Comment · Reviewer_zUE9 · 2025-11-26
> >
> > Sorry, it is my typo here. I wanted to mention about the Q1.

---

> > > ### Author Response · Authors · 2025-11-26
> > >
> > > No problem and thank you for the clarification!
> > >
> > > For Q1, we included the explanation in our reply but did not integrate it into the paper itself, since the question was noted as out of scope for the paper. Could you kindly clarify whether you’d prefer that we incorporate some of the explanation into the paper? We want to ensure we address your expectations accurately. Thanks!

---

> > > > ### Author Response · Authors · 2025-12-03
> > > >
> > > > We have added a concise version of our Q1 response to Section 3.2 of the latest revision, as it is a relevant aspect of our reward formulation.

---

### Author Response · Authors · 2025-12-03
**Summary of the Review and Discussion Process To Date**

We thank all reviewers for their careful assessments and helpful feedback. Below is a summary of the review and discussion process to date.

**Reviewers highlighted several strengths of our approach.**

* Reviewer **zUE9** noted that our method yields strong performance gains and provides explicit human-readable constraints that clarify policy behavior.
* Reviewer **EXre** emphasized the novelty of extracting a unified rule set over an entire preference dataset, the strong gains and robustness to overoptimization, and the thorough evaluations and ablations.
* Reviewer **V4rY** highlighted that our pipeline is fully automated and that the extracted rules are human-interpretable and aligned with established good practices for LLM outputs.
* Reviewer **SJ6k** commented that the rules are compact and likely to be generalizable, and noted the straightforward and concise nature of the pipeline.

**We addressed identified weaknesses with targeted clarifications and experiments.**

* Reviewer **zUE9** raised concerns about the industrial nature of the pipeline and the unclear motivation for the multi-stage design, which we addressed by clarifying that the novelty lies in systematically leveraging reasoning chains and explaining how the multi-stage structure improves stability, reduces context length, and makes each step better scoped.
* Reviewer **EXre** questioned whether gains might come from increased judge compute rather than the rules themselves, which we addressed by adding FLOP-matched experiments with long-thinking and ensemble judges that underperform AutoRule, demonstrating that improvements arise from rule structure rather than compute.
* Reviewer **V4rY** expressed concerns about the perceived marginality of gains, in-distribution effectiveness, conciseness bias, and comparisons with parallel cited work, which we addressed by emphasizing the strength of the results and our model family selections, discussing conciseness thoroughly, and including direct comparisons showing AutoRule’s advantages over RaR and RLCF.
* Reviewer **SJ6k** pointed out presentation issues, unclear notation, limited explanation of reward hacking, and areas where the pipeline could be improved. We addressed these by correcting notation, clarifying key methodology aspects, validating the connection between our results and reward hacking with supporting literature, adding a reward hacking case study, and responding thoroughly to the reviewer’s suggested improvement areas.

**We addressed the questions raised by the reviewers in detail.**
* Reviewer **zUE9** asked about dependency on the reward function and the handling of intransitivity, which we answered by explaining that AutoRule models preferences as multidimensional rules rather than collapsing them into a single scalar, and we added a concise version of this explanation to the paper.
* Reviewer **EXre** asked about how performance varies with the number of rules, which we answered by adding scaling experiments with rule set alignment accuracy and downstream model performance measures.
* Reviewer **V4rY** asked about teacher-model sensitivity, rule-set generalization, and redundancy between rules, which we answered by adding results with Qwen3-8B as the teacher model, highlighting cross-dataset improvements, and providing qualitative analysis showing that rules are largely complementary with some shared themes.
* Reviewer **SJ6k** asked about subset-size selection, alternative sampling strategies, clarification of “different rule lists,” and cross-domain alignment, which we answered by demonstrating robustness across subset sizes, discussing influence-based and diversity-based sampling, clarifying our phrasing thoroughly, and adding cross-domain alignment results.

Based on the reviewers’ feedback, **we have added new experiments, case studies, clarifications, discussions, and corrections to the updated PDF** in coordination with our responses. Specific revisions are described in our individual replies to each reviewer. We believe the revisions have strengthened the paper and thank the reviewers again for their valuable feedback.

---

### Meta-Review · Area_Chair_ocw5 · 2026-01-06

**Summary:**

This paper presents AutoRule, an automatic framework to extract rules from preference feedback and convert them to rule-based rewards. The authors demonstrated that the method achieves improvements on chat benchmarks over several baselines involving RL and variants with various types of rewards.

Overall, the reviewers found the method to be interesting and reasonable. However, some also pointed out that the method is essentially a pipeline (zUE9, SJ6k) and does not have sufficient technical depth. Reviewer V4rY also noted that the empirical results are quite marginal and weak on the benchmarks. The authors provided new results in the rebuttal using new models, but the results remained weak. Specifically, a 8B model should achieve much higher scores on benchmarks like AlpacaEval 2 even only using DPO-style preference learning, and the proposed method based on GRPO should give better results.

**Reviewer Concerns:**

The concerns regarding technical contribution and empirical significance were not adequately addressed.

**Reviewer Scores:**

The reviewer scores would unlikely go up given that the core concerns (novelty, empirical significance) remain unaddressed.

---

### Decision · Program_Chairs · 2026-01-26

Reject